# In vivo lentiviral vector gene therapy to cure hereditary tyrosinemia type 1 and prevent development of precancerous and cancerous lesions

Clara T. Nicolas[1,2,3], Caitlin J. VanLith[1], Raymond D. Hickey[1,4], Zeji Du[1], Lori G. Hillin[1], Rebekah M. Guthman[1,5], William J. Cao[1], Benjamin Haugo[1], Annika Lillegard[1], Diya Roy[1], Aditya Bhagwate[6], Daniel O'Brien[6], Jean-Pierre Kocher [6], Robert A. Kaiser[1,7], Stephen J. Russell [4] & Joseph B. Lillegard[1,7,8] ✉

Conventional therapy for hereditary tyrosinemia type-1 (HT1) with 2-(2-nitro-4-trifluoromethylbenzoyl)−1,3-cyclohexanedione (NTBC) delays and in some cases fails to prevent disease progression to liver fibrosis, liver failure, and activation of tumorigenic pathways. Here we demonstrate cure of HT1 by direct, in vivo administration of a therapeutic lentiviral vector targeting the expression of a human fumarylacetoacetate hydrolase (*FAH*) transgene in the porcine model of HT1. This therapy is well tolerated and provides stable long-term expression of FAH in pigs with HT1. Genomic integration displays a benign profile, with subsequent fibrosis and tumorigenicity gene expression patterns similar to wild-type animals as compared to NTBC-treated or diseased untreated animals. Indeed, the phenotypic and genomic data following in vivo lentiviral vector administration demonstrate comparative superiority over other therapies including ex vivo cell therapy and therefore support clinical application of this approach.

Hereditary tyrosinemia type 1 (HT1) is an autosomal recessive inborn error of metabolism that is characterized by the inability to metabolize tyrosine. Caused by a deficiency of the fumarylacetoacetate hydrolase (FAH) enzyme[1], HT1 causes an accumulation of toxic metabolites in the liver resulting in severe oxidative damage. These patients develop fibrosis, cirrhosis, high rates of hepatocellular carcinoma (HCC), and liver failure at a very young age if left untreated[2,3]. Pharmacological treatment for HT1 exists in the form of 2-(2-nitro-4-tri-fluoromethylbenzoyl)−1,3-cyclohexanedione (NTBC), a drug which inhibits the upstream enzyme 4-hydroxyphenylpyruvate dioxygenase

(HPD) thereby reducing the formation of toxic metabolites responsible for liver disease progression, thus essentially converting the disease into hereditary tyrosinemia type 3 (HT3), with a more benign phenotype[4]. For a majority of patients, NTBC greatly ameliorates the HT1 phenotype, and if initiated before 1 month of age seems to be able to reduce the development of HCC in at least the first three decades of life[5]. However, it remains to be seen if the prevention of HCC persists during adult years, as introduction of this drug took place during the 1990s[6]. A number of case reports and series point to the development of fibrosis, cirrhosis, and HCC despite NTBC therapy, especially when

[1]Department of Surgery, Mayo Clinic, Rochester, MN, USA. [2]Faculty of Medicine, University of Barcelona, Barcelona, Spain. [3]Department of Surgery, University of Alabama Birmingham, Birmingham, AL, USA. [4]Department of Molecular Medicine, Mayo Clinic, Rochester, MN, USA. [5]Medical College of Wisconsin, Wausau, WI, USA. [6]Department of Biomedical Statistics and Informatics, Mayo Clinic, Rochester, MN, USA. [7]Midwest Fetal Care Center, Children's Hospitals and Clinics of Minnesota, Minneapolis, MN, USA. [8]Pediatric Surgical Associates, Minneapolis, MN, USA. ✉e-mail: jlillegard@msn.com

NTBC therapy was delayed[7–10]. The frequency of HCC within the limited follow-up period available for patients treated with NTBC appears to be <1% if started prior to 1 year of age, 7% if started between 1 and 2 years of age, 21% if started between 2 and 7 years of age, and 35% if started after 7 years of age[11]. Therefore, these patients require lifelong HCC surveillance with alpha-fetoprotein (AFP) levels drawn every 3–6 months[6]. In addition, HT1 patients treated with NTBC show progressive neurocognitive decline despite optimal NTBC dosing, possibly due to prolonged exposure to elevated tyrosine levels that occur on the drug[12–14]. Although the exact pathophysiological mechanisms behind these deficits are not clear[6], two studies have identified a correlation between the phenylalanine to tyrosine ratio in these patients and IQ[15,16]. This theory is further supported by the fact that HT3, the disease caused by a deficiency in the same enzyme that NTBC blocks, also presents with neurological manifestations including developmental delay[17–20]. Complications attributed to NTBC treatment other than low IQ include attention deficits, memory and processing problems, psychomotor and behavioral issues, and impairment of executive function and social cognition[21–28]. With daily dosing of NTBC, as well as frequent testing, many patients struggle with compliance issues related to NTBC therapy in the second and third decades of life[26,27]. Non-compliance with NTBC therapy is a key factor in disease progression for HT1 patients[29]. To date, the only therapeutic option for patients refractory to NTBC or with disease progression despite NTBC therapy, and the only curative treatment for HT1, is liver transplantation.

Gene therapy may offer HT1 patients an alternative, less invasive cure. Our group has previously demonstrated that ex vivo lentiviral (LV) gene transfer followed by autologous transplantation of corrected hepatocytes is curative in both mouse and pig models of HT1[30]. The ex vivo approach is effective, but autologous transplantation requires a partial hepatectomy in liver-based diseases to avoid the immune rejection that would occur with transplantation of allogeneic hepatocytes. Therefore, in vivo liver-directed gene therapy presents an attractive non-surgical alternative for the treatment of inborn errors of metabolism of the liver.

Limited biochemical correction of the mouse model of HT1 has been achieved in vivo through retrovirus, adenovirus, and adeno-associated virus (AAV) mediated gene transfer[31–33], as well as through RNA-guided CRISPR/Cas9-mediated genome editing and metabolic pathway reprogramming[34,35]. Thus far, none of these in vivo techniques have been tested in the more clinically relevant large animal model of HT1. The mouse model does not recapitulate some of the most critical aspects of the human phenotype as mice develop HCC independently of fibrosis and cirrhosis, while pigs and humans do not[36]. In mice, HCC develops despite long-term NTBC treatment, and it does so in the setting of hepatocyte dysplasia, liver steatosis, anisocytosis, and lymphocytic infiltrates, but not widespread fibrosis and cirrhosis, as it does in human HT1 patients[37]. It is well known at this point that mice develop HCC through an entirely different set of molecular mechanisms when compared to pig and human, with differences including but not limited to: a basal metabolic rate in mice about seven times higher than in humans, affecting the levels of endogenous oxidants and other mutagens that are produced as by-products of normal oxidative metabolism; a difference in the way that many carcinogens are activated or neutralized; a proclivity towards the development of mesenchymal tissue cancers instead of the epithelial-cell layer carcinomas more common in humans; a high frequency of spontaneous cell immortalization in mice, possibly explained by their constitutive expression of telomerase, which is not usually expressed in human cells; a difference between the two species in the relative contributions of the p53 and retinoblastoma pathways to cell senescence; and key differences in the signaling requirements for cell transformation, where in mice perturbation of just the p53 and Raf-MAPK pathways

seems to be sufficient to mediate tumorigenic conversion, while in human fibroblasts perturbation of six or more pathways is necessary to achieve the same outcome[38,39]. In addition, the efficiency of AAV-based therapy in humans is hampered by pre-existing neutralizing antibodies to the viral vector that develop after natural exposure to the wild-type virus and, in the case of liver disease, by immune-mediated clearance of AAV-transduced hepatocytes[40]. The same is true of the immune response to the bacterial Cas9[41]. AAV-based systems have been used in adult patients with hemophilia and other diseases but these therapies would not be beneficial in HT1 patients as liver injury with subsequent hepatocyte death and regeneration leads to the loss of transgene expression in hepatocytes over time[42–44].

In the human setting, HIV-derived LV vectors hold several potential benefits over AAV vectors. First, the low prevalence of HIV infection in humans decreases the chances of an immune response curtailing the therapy's effectiveness[45]. Second, LV vectors are better suited for the treatment of certain diseases due to their ability to carry larger gene inserts[46]. Finally, in HT1 patients and the pediatric population, where hepatocyte turnover is high, the use of integrating LV vectors may be key for stable, lifelong expression of the desired transgene, whereas episomal vectors will be progressively diluted out during cell replication until effectively lost[47]. This advantage is amplified in HT1 as FAH-positive hepatocytes are selected for at the expense of injured FAH-negative hepatocytes.

Liver-based LV gene therapy has yielded promising results in a large animal model of hemophilia B; Cantore et al. showed that intraportal administration of a LV vector carrying the cFIX transgene is able to safely provide stable clinical benefit in dogs with this disease, and in doing so provided preclinical proof-of-concept for the in vivo use of LV vectors in liver gene therapy in their hemophilia model[48]. In this study, we evaluated both the effectiveness and safety profile of in vivo liver-directed LV gene therapy for the treatment of a pro-cancerous metabolic disease, HT1.

## Results

### In vivo portal vein delivery of a LV vector is well tolerated in pigs

To characterize the acute effects of LV vector administration in pigs and determine the optimal route of administration, we initially delivered a LV carrying the human *FAH* transgene expressed with the liver-specific alpha-1-antitrypsin (AAT) promoter (LV-FAH, Fig. 1a) systemically via an ear vein. We found no significant differences in key cytokines, including TNF-alpha and IL-6, or white blood cell count response, between this animal and our reference lentiviral vector with green fluorescent protein when administered via portal vein (LV-GFP, Fig. 1b, Supplementary Fig. 1a). In both cases, an acute increase in inflammatory markers was observed after administration, with peak concentrations being reached at approximately 1 h after the end of injection, and a subsequent decrease to pre-treatment levels after 6–12 h. Circulating white blood cell counts decreased minimally following administration of the LV vector, and recovered to their predose range within hours. However, despite pre-treatment with an established immunosuppression protocol[48], systemic administration resulted in an anaphylactoid reaction with acute hypotension (mean pressure of 48 mmHg) that required vasopressor support (0.2 mg of epinephrine IV). To ensure this reaction was not related to the LV-FAH construct, we delivered LV-GFP systemically to another animal, and again observed a similar reaction to infusion of the vector (mean pressure 42 mmHg), requiring support for maintenance of vital functions (IV fluid bolus, 0.2 mg of epinephrine IV and 1 mg of atropine IV, Supplementary Fig. 1c).

Based on these results, we pursued a minimally invasive approach using percutaneous ultrasound-guided portal vein administration of the LV vector carrying the human FAH transgene in four *FAH*[−/−] pigs (Table 1). Doses were determined based on previous in vivo LV liver-

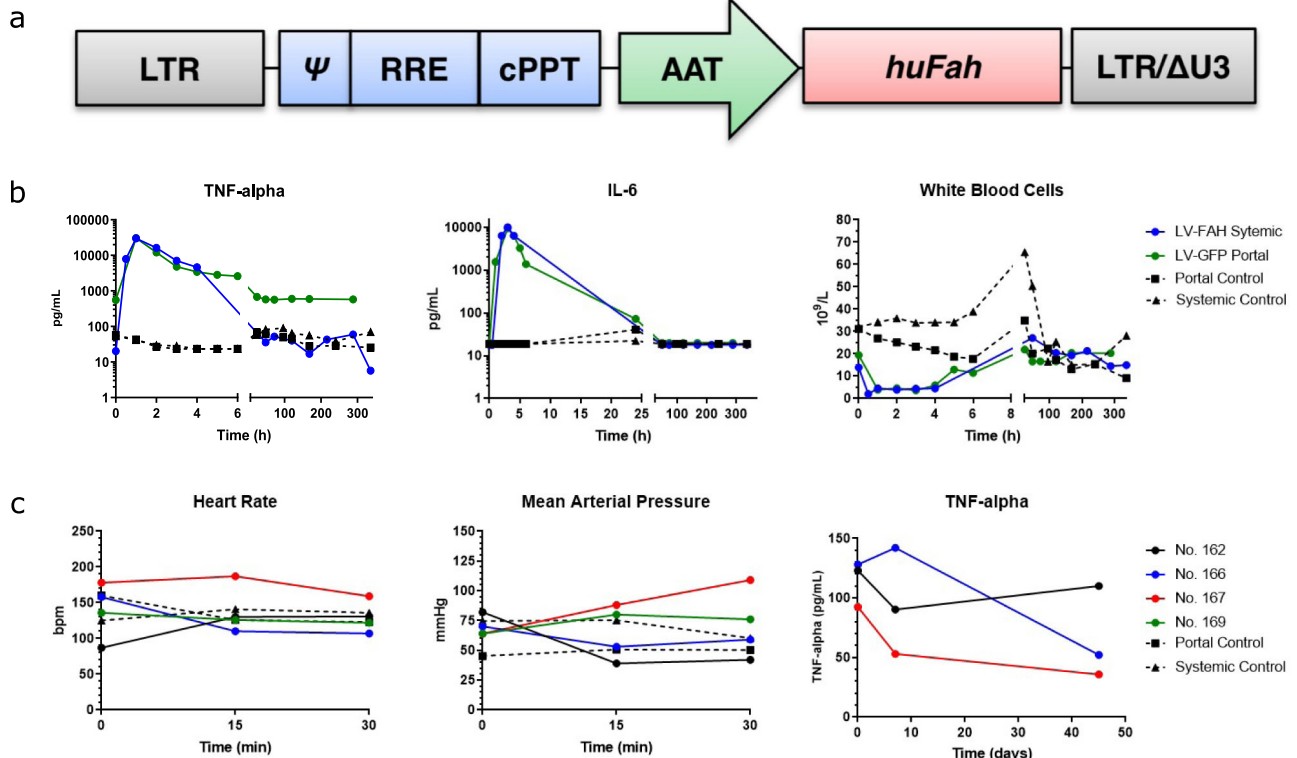

**Fig. 1 | LV-FAH administration to pigs. a** Schematic representation of the lentiviral vector carrying the human *FAH* cDNA under control of the human alpha-1-antitrypsin (AAT) promoter. LTR long terminal repeat, Ψ psi packaging sequence, RRE Rev-responsive element, cPPT central polypurine tract; LTR/ΔU3, 3′ long terminal repeat with deletion in U3 region. **b** Acute inflammatory response after systemic LV-FAH administration compared to reference portal vein LV-GFP delivery. No significant differences in TNF-alpha or IL-6 concentrations or white blood cell count were found between the two routes of administration. Note that 30,000 pg/mL represents the upper limit of detection for the TNF-alpha assay. **c** Vital signs and TNF-alpha levels during and after portal vein delivery of LV-FAH in four *FAH*−/− animals. Heart rate and blood pressure remained stable throughout infusion, and TNF-alpha levels were within normal range immediately post-infusion and at 8 and 45 days post-treatment.

### Table 1 | Gene therapy dose in treated pigs

| Pig | Genotype | Sex | Age (weeks) | Weight (kg) | Vector | Route | Dose (mL) | Dose (TU/kg) | Study duration |
|---|---|---|---|---|---|---|---|---|---|
| No. 169 (Carmen) | *FAH*−/− | F | 6 | 11.8 | LV-FAH | Portal | 150 | $2.00 \times 10^{10}$ | 2 days[a] |
| No. 162 (Santiago) | *FAH*−/− | M | 6 | 10.4 | LV-FAH | Portal | 135 | $2.04 \times 10^{10}$ | 60 days |
| No. 166 (Pepe) | *FAH*−/− | M | 6 | 10.8 | LV-FAH | Portal | 140 | $2.04 \times 10^{10}$ | 337 days |
| No. 167 (Lola) | *FAH*−/− | F | 6 | 13.6 | LV-FAH | Portal | 170 | $1.96 \times 10^{10}$ | 337 days |
| No. 762 | *FAH*−/− | M | 6 | 10.2 | LV-FAH | Systemic | 224 | $2.00 \times 10^{10}$ | 14 days |
| No. 764 | *FAH*+/− | M | 6 | 9.2 | LV-GFP | Portal | 305 | $2.00 \times 10^{10}$ | 13 days |

[a]Animal recovered from anesthesia slowly and was found dead on Day 2 due to a suspected inflammatory reaction; TU transducing units.
Four animals (two male and two female) were assigned to dosing as indicated below. Animals were surgically anesthetized and dosed with $2 \times 10^{10}$ TU/kg LV-FAH directly via the portal vein.

directed gene therapy work in large animals[48]. Animals that received portal vein delivery of the LV-FAH construct tolerated the infusion well with no acute complications and maintained stable vital signs throughout the procedure (Fig. 1c, panels 1 and 2, Supplementary Fig. 1b). However, Pig No. 169 recovered from anesthesia and the infusion event but died acutely on day 2 post-treatment with no significant prodrome before death. Autopsy revealed no significant liver pathology or systemic inflammatory disease including pulmonary edema or microhemorrhage. However, this animal was found to have severe hypertrophic cardiomyopathy (Supplementary Fig. 2), which may have contributed to its sudden demise. Hypertrophic cardiomyopathy is a known complication in Landrace pigs with prevalence of up to 23% in some studies, and spontaneous sudden mortality of up to 18%, particularly related to stress-inducing situations such as transportation[49,50]. In the remaining animals, neither early expression

of the transgene (8 days) nor expansion of the corrected cells (45 days) had an effect on TNF-alpha levels (Fig. 1c, panel 3), which remained within normal range.

### In vivo LV-FAH delivery cures the pig model of HT1 with no residual liver pathology

NTBC administration was discontinued at the time of vector infusion to stimulate expansion of newly transduced, FAH-positive hepatocytes. The three remaining animals were then cycled on and off NTBC based on clinical and weight parameters until they were able to maintain a healthy, NTBC-independent growth curve. Pig No. 166 achieved NTBC-independent growth at day 98 post treatment and Pig No. 167 at day 78 post-treatment, both after four cycles on the drug (Fig. 2a). At the time of planned euthanasia (1 year old, 337 days post-treatment), these pigs had been off NTBC for 239 and 259 days

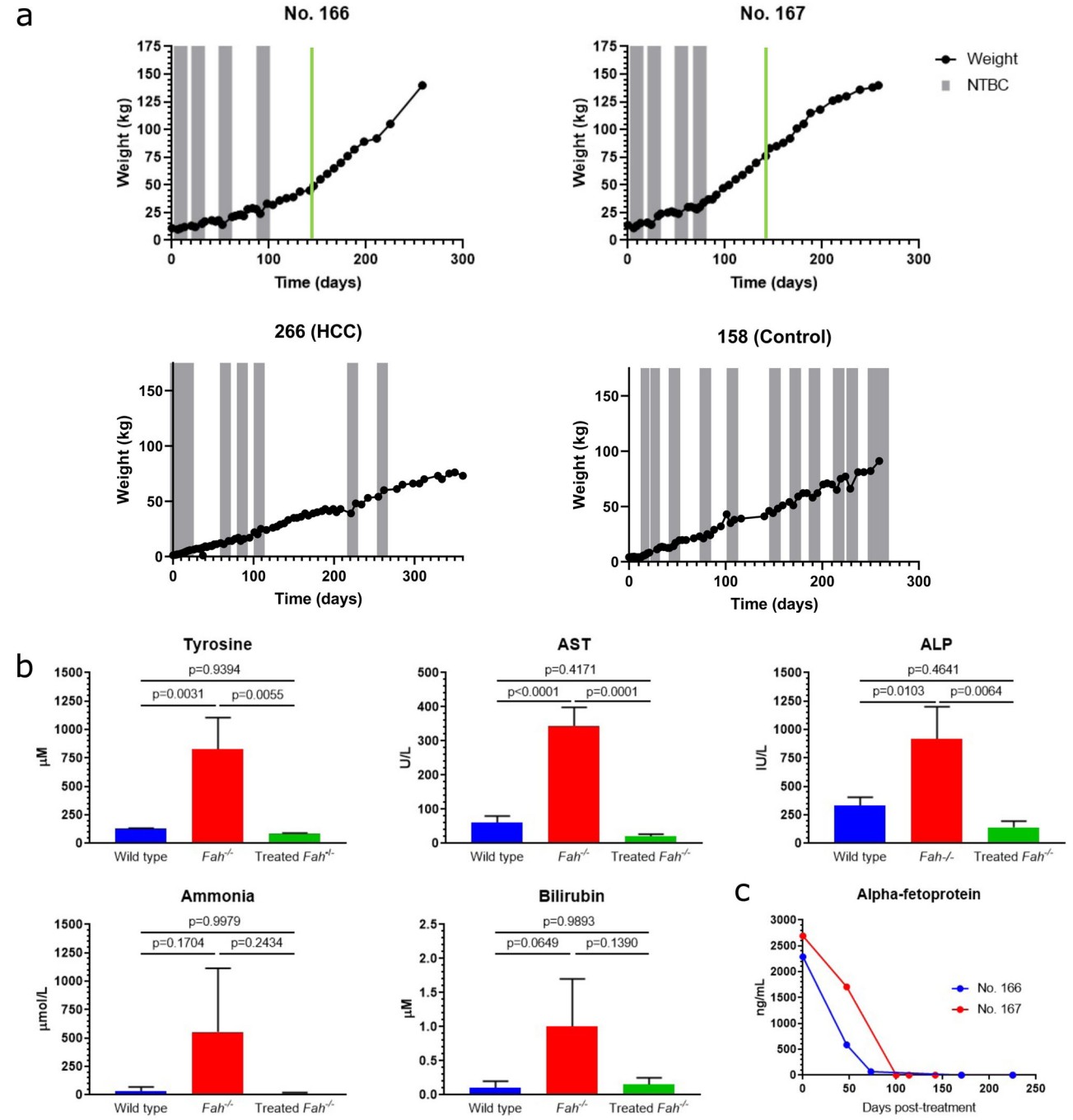

**Fig. 2 | NTBC-independent weight gain and correction of biochemical parameters after in vivo LV-FAH delivery. a** Timeline of NTBC cycling and withdrawal for No. 166 and No. 167 (first row) demonstrated NTBC-independent growth at 98 and 78 days post-treatment. Each gray bar represents 1 week spent on NTBC. Green line represents when blood was drawn for the biochemical data in (**b**). Timeline of NTBC undertreatment for No. 266, which did not receive LV-FAH and developed HCC, and for No. 158, which did not receive LV-FAH but received appropriate NTBC cycling. **b** Normalization of tyrosine, aspartate aminotransferase (AST), alkaline phosphatase (ALP), ammonia, and total bilirubin levels at 142 days post-treatment.

Mean ± standard deviation are shown for No. 166 and No. 167 alongside historical wild-type ($n = 2$) and untreated $FAH^{-/-}$ ($n = 3$) controls. *P*-values are provided for treated animals compared to untreated $FAH^{-/-}$ controls based on a two-sided Welch's *t*-test with no adjustment for multiple comparisons: * represents $P < 0.05$, ** represents $P < 0.01$, **** represents $P < 0.0001$. Tyrosine: $p < 0.0001$; AST: $p = 0.01$; ALP: $p = 0.004$; ammonia: $p = 0.03$; bilirubin: $p = 0.04$. **c** Age-appropriate decreases and maintenance of minimal alpha-fetoprotein (AFP) levels in No. 166 and No. 167 despite chronic NTBC withdrawal.

respectively with no health issues, and with and average weight gain of between 0.6 and 0.9 kg per day, which is normal for healthy, wild-type Large White and Landrace pigs[51,52].

Biochemical parameters, including tyrosine levels and liver function tests, were measured to confirm correction of the disease phenotype. At 142 days post-treatment, tyrosine levels were 79 μM in No.

166 and 63 μM in No. 167. These levels are within normal limits for a wild-type pig and significantly lower than those of historical $FAH^{-/-}$ controls ($p = 0.04$, Fig. 2b). Normal tyrosine levels were maintained throughout all successive blood draws until the time of euthanasia (Supplementary Fig. 3). Aspartate aminotransferase (AST), alkaline phosphatase (ALP), ammonia, and bilirubin levels were also

normalized at day 142 post-treatment (Fig. 2b). All measured liver function tests were comparable to wild-type levels at this time point, despite animals having been off NTBC for 45–65 days. In fact, all liver function tests remained within normal limits throughout the experiment with the exception of ALP in both animals and gamma-glutamyl transferase (GGT) in one animal, which were elevated at only one time point (47 days post-treatment), between the third and fourth cycles on NTBC (Supplementary Table 1). Additionally, AFP levels showed an age-appropriate decrease over time in both animals, remaining stable at 0.25 ng/mL at the end of the study (Fig. 2c) which is identical to wild-type age matched controls[53].

To allow for interim analysis, Pig No. 162 was euthanized at day 60 post-treatment after three cycles of NTBC. After harvesting the complete liver from No. 162, the presence of FAH-positive hepatocytes was evaluated. After examining over a hundred sections from the right, middle, and left lobes of the liver we found an even distribution of clonally expanded FAH-positive cells were present, constituting approximately 10% of the total liver cells (Fig. 3b, top row). No active inflammation, fibrosis or necrosis was found in routine histology using H&E or Masson's Trichrome staining (Fig. 3b, top row). In contrast, no FAH-positive cells or liver abnormalities were found in Pig No. 169 48 h post-treatment (Supplementary Fig. 4). The longer-term animals (Nos. 166 and 167) both underwent laparoscopic liver biopsies from the right and left lobes of the liver at 225 days post-treatment. Here, extensive liver repopulation with FAH-positive hepatocytes had occurred, totaling 69% and 78% of total number of cells in the liver, respectively (Fig. 3b, second and third row). As previously described after liver repopulation in $FAH^{-/-}$ animals, individual hepatocyte clones were uniquely identified by variations in FAH-staining intensity between and within lobules[30]. At this point, multiple abnormalities were seen in 50 to 80% of the liver of both animals, including inflammation and reversible fibrosis that could be compared to Stage 2 liver fibrosis on the porcine-adapted METAVIR scoring system[54,55], as well as hepatocyte inflammation and minimal bile stasis (Fig. 3b, second and third row). These findings did not correlate with increased AFP staining. At 337 days post-treatment, samples were taken from the right, middle, and left lobes and after examining over 100 sections both animals showed near complete liver repopulation with FAH-positive hepatocytes (Fig. 3a, b, fourth and fifth row) with near total resolution of the fibrosis and inflammation seen at day 225 along with restoration of normal hepatic architecture resembling wild-type liver (Supplementary Fig. 5a–c). H&E/Trichrome sections showed normal hepatocyte morphology and minimal fibrosis in 10–15% of the liver, which was confined to the periportal hepatocytes (Fig. 3b, fourth and fifth row). Full quantification of fibrosis and METAVIR score for each experimental liver section analyzed is provided in Supplementary Table 2 and gross histology images are provided in Supplementary Fig. 5d–f. Importantly, adult age-matched wild-type animals ($n = 2$) show uniform hepatocellular morphology, with similar fibrosis limited to the connective tissue between lobules and periportal hepatocytes (porcine-adapted METAVIR score 0–1), and negligible AFP staining (Fig. 3b, left column). Untreated or chronically undertreated $FAH^{-/-}$ adult pigs (No. 266, prior experiment, Fig. 2a) develop cirrhosis, adenomas, and ultimately HCC at 12 months, as shown here with hepatocellular morphological derangement, extensive stage 4 fibrosis, and significant AFP staining (Fig. 3c, right column). In the LV-FAH treated animals, no such lesions were present grossly after sectioning the entire organ into one centimeter slices, and none of these pathological features were present in any sections evaluated histologically. Untreated Pig No. 266 showed appropriately elevated AFP at birth similar to Nos. 166 and 167 (Fig. 2c) and to wild-type controls; however, at the time of necropsy (1 year old) No. 266 showed increased AFP levels to 168 ng/mL, further supporting the development of HCC in this animal as compared to Nos. 166 and 167, whose AFP levels at

the time of necropsy are below 0.5 ng/mL (Fig. 3d). In addition, Ki-67 and TUNEL staining were performed to assess levels of hepatocyte proliferation and apoptosis (Supplementary Fig. 6). While Ki-67 staining expectedly increased during the process of liver repopulation with FAH-positive hepatocytes, it was at wild-type levels by 337 days post-treatment, mirroring levels of inflammation and fibrosis seen on routine histology.

## Portal vein delivery of a LV vector is effective at targeting the liver with limited off-target bio-distribution

PCR screening of multiple tissue types was performed after portal and systemic LV-FAH delivery to evaluate for presence of the LV vector in target and non-target tissues. At 48 h post portal vein delivery (No. 169) positive vector presence was limited to all lobes of the liver (Fig. 4a, top left). At 60 days (No. 162), evidence of vector integration was present only in the liver as well (Fig. 4a, top right). Finally, at 337 days (Nos. 166 and 167) there was strong evidence of vector integration again in all lobes of the liver in both animals, with no evidence of off-target lentiviral integration elsewhere (Fig. 4a, middle). In contrast, the pig in which LV-FAH was delivered systemically via ear vein injection showed no evidence of vector presence in the liver by standard PCR, whereas spleen, pancreas, duodenum, and lung were positive for LV-FAH at 14 days post-infusion (Fig. 4a, bottom). At 60 days post-treatment, LV integration in the liver was measured at under one vector copy per genome (VCG), with this number increasing in long-term animals as FAH-positive hepatocytes expanded to repopulate the liver (Fig. 4h).

## LV-FAH vector shows a benign integration profile in pig hepatocytes after in vivo delivery

In order to characterize the LV-FAH vector integration profile in hepatocytes after in vivo delivery, we performed next-generation sequencing and bioinformatics analysis of hepatocytes at the time of euthanasia for animals at 60 (No. 162) and 337 days (Nos. 166 and 167, Fig. 4b–h). Mapping statistics are provided in Supplementary Table 3.

Integration sites were present across the genome in all animals (Fig. 4b). There were no significant differences in general LV integration profile in the liver between the two long-term animals; however, integration profile in the 60-day animal (No. 162) showed some unique characteristics. Interestingly, at 60 days, relative integration frequency was higher in exons as compared to introns, while at 337 days (Nos. 166 and 167) integration was more frequent in introns than in exons (Fig. 4c), possibly indicating preferential propagation of cells with intronic integrations over the course of liver repopulation. Relative LV integration frequency remained similar across gene expression levels (Fig. 4d). At both time points (all three animals), LV integration was very rare in CpG-rich promoter regions relative to their percentage of the genome as compared to non-CpG islands ($p = 0.0001$, Fig. 4e), with a trend toward non-tumor coding genes for the longer-term animals (Fig. 4f), possibly again indicating a selective preference for those hepatocytes over time. Lastly, integration was most prevalent downstream of transcription start sites, suggesting unlikely activation of any oncogenes from proximal integration of the AAT promoter-containing construct (Fig. 4g). The top ten most common genes in terms of integration frequency for each animal and tissue type are shown in Table 2. LV-FAH integration was also found to be increased at the $FAH$ and $SERPINA1$ loci suggesting a role for sequency homology in integration preference. Gene Set Enrichment Analysis (GSEA) demonstrated enrichment of the PI3K-Akt and TGFb signaling pathways only in No. 167, while No. 166 demonstrated no hits (Fig. 4i). There were no changes in gene expression in any tumor-related genes.

Finally, RNAseq demonstrated expression profile similarities between wild-type, LV-FAH treated, and NTBC-treated pigs as compared to tumor-positive pig liver, particularly in cancer-related and fibrosis-related genes (Fig. 5a). Interestingly, while LV-FAH treated

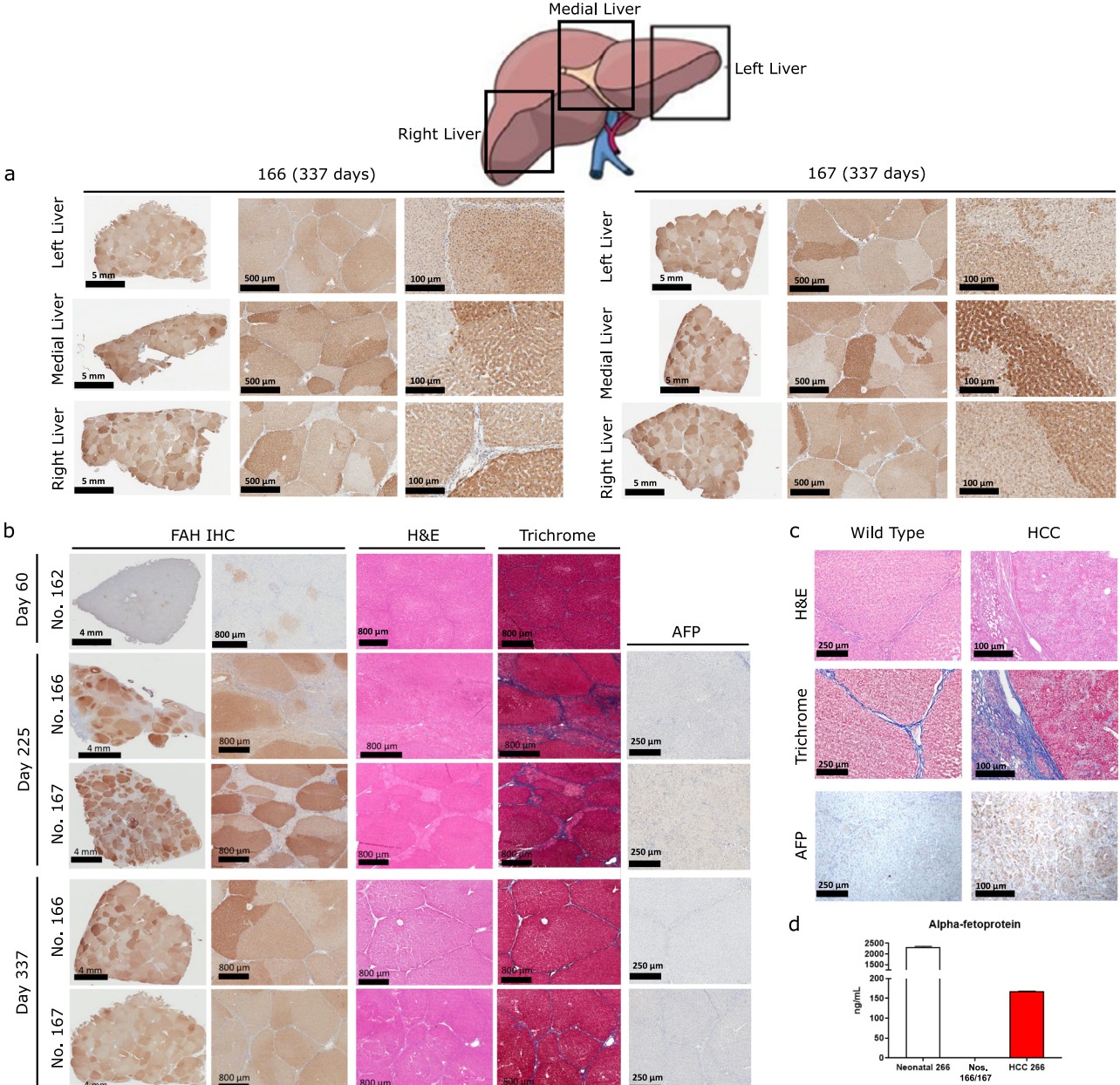

**Fig. 3 | Liver repopulation with FAH-positive hepatocytes after in vivo LV-FAH delivery. a** Immunohistochemistry (IHC) for FAH in Nos. 166 and 167 at the time of euthanasia (337 days post-treatment) showing liver repopulation with FAH-positive hepatocytes in all areas of the liver. Sixteen areas of each liver were examined with similar results for **a–c. b** IHC for FAH in No. 162 at 60 days post-treatment showing multiple FAH-positive nodules in the liver with no presence of significant liver injury on H&E or Trichrome staining (first row). IHC for FAH from laparoscopic liver biopsy in No. 166 (second row) and No. 167 (third row) at 225 days post-treatment showing extensive liver repopulation with FAH-positive hepatocytes, with H&E and Trichrome staining showing hepatocellular inflammation with moderate fibrosis but no increased AFP staining. IHC for FAH in No. 166 (left medial lobe, fourth row) and No. 167 (left lateral lobe, fifth row) at 337 days post-treatment showing near complete liver repopulation with FAH-positive hepatocytes, with H&E and Trichrome staining showing resolution of hepatocellular inflammation and fibrosis with no increased AFP staining. **c** H&E and Trichrome staining of wild-type pig liver (left panels) and HCC in an NTBC under-treated *FAH*⁻/⁻ pig liver (No. 266, right panels) at 1 year show liver fibrosis and cellular alterations in the under-treated animal, while IHC for AFP shows ubiquitous increases in expression consistent with development of HCC in this context. **d** Quantification of AFP in No. 266 shows age-appropriate AFP levels in the neonatal period; however, AFP was slightly elevated at 1 year, while AFP in Nos. 166 and 167 at 337 days post-treatment was below the limit of quantitation.

animals at 6 months showed an expression profile similar to both *FAH*⁻/⁻ sick or untreated animals (Fig. 5b, top) and NTBC-sustained *FAH*⁻/⁻ animals (Fig. 5b, bottom), by 12 months post-treatment where liver repopulation with healthy, FAH-positive hepatocytes had reached 100%, the LV-FAH group is most similar to the wild-type expression profile. When gene expression was compared to 6 month and 12-month time points for our LV-FAH treated animals we found gene expression differences clustered into disease pathways of importance in the progression of tyrosinemia, including general tumorigenicity, inflammation and fibrosis and abdominal and digestive cancer (Fig. 5c). We also performed a principal component analysis including all genes (*n* = 1260) from quartile 4, i.e., genes that met our filtering criteria with the highest overall level of expression when divided into quartiles, further supporting the conclusion that transcriptome-wide the wild-type pigs are more similar to the LV-FAH treated pigs than the NTBC-treated pigs (Fig. 5d).

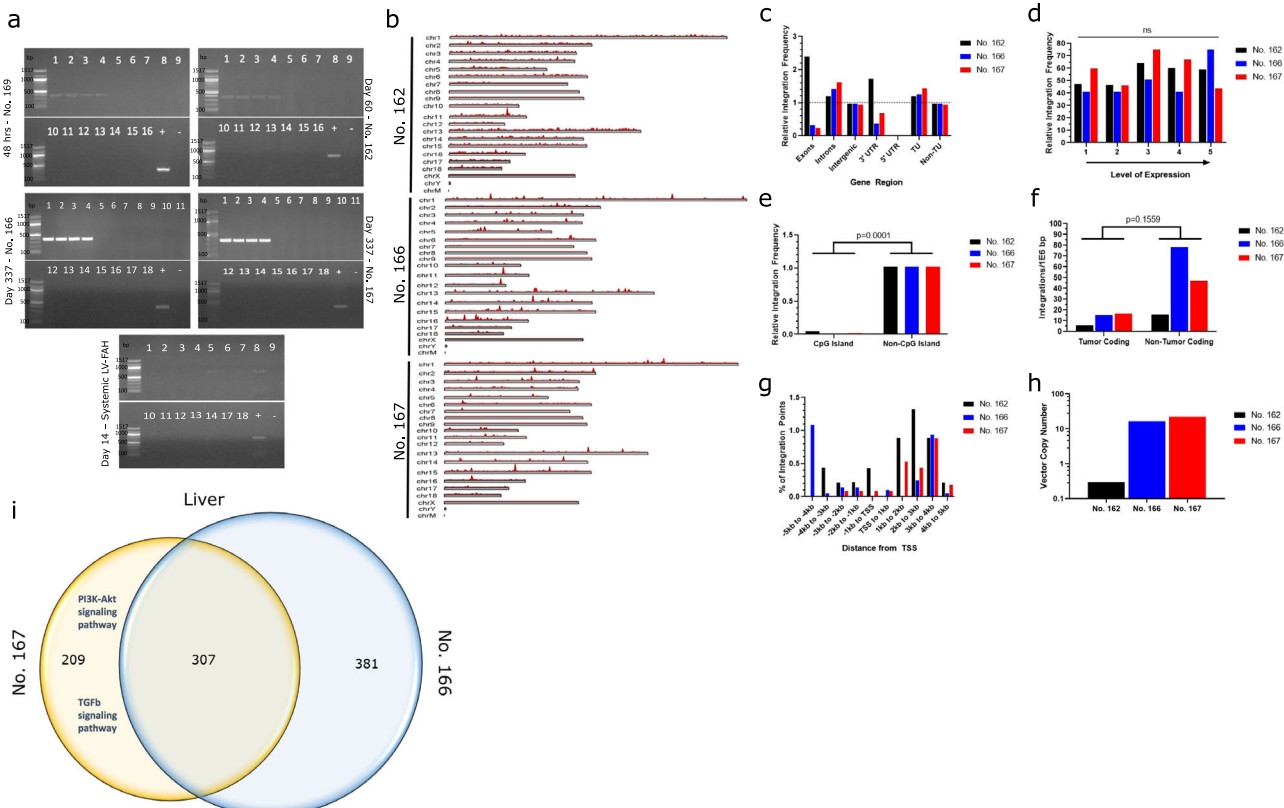

**Fig. 4 | LV-huFAH shows a benign integration profile in pig hepatocytes after in vivo exposure portal vein delivery. a** Biodistribution of LV-FAH as determined by selective PCR amplification of long terminal repeats over 48 h, 60 days, and 337 days in animals that received portal vein (No. 169, No. 162, No. 166, No. 167, respectively) and systemic injection of LV-FAH. Numbers represent the following tissues: (1) right lateral liver, (2) right medial liver, (3) left lateral liver, (4) left medial liver, (5) spleen, (6) kidney, (7) pancreas, (8) duodenum, (9) jejunum, (10) ileum, (11) colon, (12) mesenteric lymph node, (13) heart, (14) right lung, (15) left lung, (16) mediastinal lymph node, (17) testes or ovaries, (18) brain. This experiment was repeated with three primer pairs with similar results. **b** Chromosomal maps of integration events found in No. 162, No. 166, and No. 167 livers. **c** Relative LV

integration across exons, introns, and intragenic regions. **d** Relative LV integration frequency across the genome as stratified by expression level. *P*-values for d-f are based on a two-tailed z-test with no adjustment for multiple comparisons. **e** Relative LV integration frequency showing preference for non-CpG island sites, when adjusted for prevalence in the genome. **f** Relative LV integration frequency showing no preference tumor-coding genes. **g** Relative integration frequency as stratified by distance from transcription start site. **h** Lentiviral vector copy number found in liver at the time of necropsy. **i** Comparison of LV-FAH integration sites in the livers of No. 167 and No. 166 show high similarity, with 307 common integrations. Gene Set Enrichment Analysis (GSEA) demonstrated no hits in No. 166, but enrichment of the PI3K-Akt and TGFb signaling pathways in No. 167.

## Discussion

In this study, we demonstrated that lentiviral vector gene therapy is safe and effective for direct in vivo percutaneous portal vein administration in the relevant preclinical large animal model of HT1. Remarkably, the safety findings in this experiment support the in vivo application of this integrating vector with the absence of promotor- or transgene-specific adverse events and in the setting of complete resolution of the biochemical and histologic features of tyrosinemia that lead to progressive systemic and liver disease. Experimental *FAH*[-/-] pigs were cycled on and off NTBC for selection of FAH-positive hepatocytes and were cured of their disease after a single intraportal dose of the LV vector carrying the human *FAH* under control of the liver-specific HCR-AAT promoter.

Percutaneous portal vein infusion of LV-FAH was well tolerated, with no systemic inflammatory reaction. Vitals were closely monitored for 48 h after administration of the vector and showed minimal changes in heart rate and blood pressure with stable temperatures. When the same vector was delivered systemically, however, animals developed an acute hypotensive anaphylactoid-like reaction requiring vasopressor support, likely related to a transient rise in circulating cytokines TNF-α, IL-6, IL-8 and INF after exposure to LV particles[56], but returning to baseline by 6–12 h post-injection. Naldini and colleagues have previously described similar results after an open surgical

approach with portal vein administration of LV carrying a factor IX transgene in dogs[48]. This inflammatory response has been seen previously in mice exposed to LV particles, thought to be related to activation of hepatosplenic plasmacytoid dendritic cells triggered by contaminating nucleic acids or by the presence of the LV particles themselves[56]. These results could be avoided or reduced by optimization of pre-existing immunosuppression protocols[48], by species-specific dose titering to minimize systemic exposure, and through the use of the percutaneous delivery methods described in this study. Percutaneous portal vein cannulation is an advanced but obtainable skill that would allow for liver-directed gene therapy in humans to be administered under conscious sedation as opposed to general anesthesia, therefore avoiding the confounder and added physiologic stress of undergoing an open operation resulting in activation of significant inflammatory pathways seen with surgery.

Percutaneous portal vein delivery was also demonstrated to be very effective at targeting the liver in contrast to systemic delivery, limiting biodistribution to off-target tissues. Extensive PCR analysis showed highly restricted extrahepatic tissue targeting when compared to systemic administration. Although examination of biodistribution after systemic delivery was only performed in one animal at 14 days post-treatment, biodistribution results in the portal vein delivery group were consistent between all four animals at all time points,

**Table 2 | Top 10 genes by integration frequency**

|  | Gene | Number of integration sites | Percent of integration sites | Number of reads | Percent of reads | FAH homology (nucleotides/percent) |
|---|---|---|---|---|---|---|
| No. 162 | ZNF24 | 12 | 1.319 | 29698 | 29.991 | 11/100% |
|  | MIR17 | 14 | 1.538 | 17995 | 18.172 | 7/100% |
|  | MIR92A-1 | 4 | 0.440 | 4128 | 4.169 | none |
|  | TRPC4 | 5 | 0.549 | 3088 | 3.118 | 11/100% |
|  | TTR | 3 | 0.330 | 2266 | 2.288 | 12/92% |
|  | PRMT6 | 7 | 0.769 | 1954 | 1.973 | 12/92% |
|  | HSP90AB1 | 1 | 0.110 | 1663 | 1.679 | 11/100% |
|  | FAM107A | 4 | 0.440 | 1290 | 1.303 | 24/83% |
|  | NCOA1 | 2 | 0.220 | 1231 | 1.243 | 11/100% |
|  | IFI44 | 3 | 0.330 | 1202 | 1.214 | 17/82% |
| No. 166 | ACVR2A | 12 | 0.288 | 468183 | 8.104 | 9/100% |
|  | VGLL3 | 46 | 1.103 | 346816 | 6.003 | 10/100% |
|  | UOX | 27 | 0.647 | 282277 | 4.886 | 9/100% |
|  | MYC | 4 | 0.096 | 240240 | 4.158 | 13/92% |
|  | ITGA2 | 98 | 2.350 | 189860 | 3.286 | 11/100% |
|  | MIR7135 | 14 | 0.336 | 178740 | 3.094 | 28/82% |
|  | ZRANB2 | 77 | 1.847 | 166959 | 2.890 | 39/74% |
|  | AKTIP | 13 | 0.312 | 163213 | 2.825 | 11/100% |
|  | PNPT1 | 17 | 0.408 | 159257 | 2.757 | 13/92% |
|  | MGAT4C | 24 | 0.576 | 100552 | 1.740 | 17/88% |
| No. 167 | DPP4 | 53 | 2.340 | 93939 | 3.649 | 14/93% |
|  | ASB3 | 31 | 1.369 | 83777 | 3.255 | 14/86% |
|  | SH3GLB1 | 6 | 0.265 | 78216 | 3.038 | 19/79% |
|  | DPH5 | 4 | 0.177 | 78209 | 3.038 | 16/94% |
|  | RSPO3 | 8 | 0.353 | 77394 | 3.007 | 10/100% |
|  | ITGAV | 6 | 0.265 | 75559 | 2.935 | 15/87% |
|  | CTSB | 22 | 0.971 | 59593 | 2.315 | 18/83% |
|  | CCL28 | 50 | 2.208 | 55120 | 2.141 | 11/100% |
|  | DDX58 | 21 | 0.927 | 54736 | 2.126 | 21/81% |
|  | MGAT4C | 5 | 0.221 | 52320 | 2.032 | 17/88% |

LV-FAH integration sites were evaluated from representative sections of liver from pigs Nos. 162, 166, and 167. Data are reported by the percent of reads each individual gene represented of the total for that animal/tissue. The number of unique integration sites within a given gene, and the percent of unique integrations sites that represents of the total for each tissue/animal is also presented for context. Finally, the region of highest homology to *FAH* is presented for each. MGAT4C, included in the top ten genes with highest relative integration frequency for both pigs Nos. 166 and 167, is a transferase essential to production of some N-linked sugar chains.

including early (48 h), mid (60 days), and late (337 days). Initial lentiviral integration into the liver was measured at less than one VCG, well below FDA recommendations of five VCG[57]. This number rapidly increased to ten in our long-term animals as corrected hepatocytes expanded to repopulate the liver, possibly suggesting natural selection of hepatocytes with higher VCG or more robust enzymatic activity.

Importantly, liver-directed in vivo LV gene therapy cured HT1 in our experimental animals and did so in a more rapid timeframe than ex vivo gene therapy using the same vector construct, therefore offering less opportunity for liver damage. Two long-term pigs demonstrated NTBC independence at day 78–98 post-treatment, after only four cycles on the drug. By comparison, our previous ex vivo study resulted in NTBC independence at 93–150 days post-treatment, after four to six cycles on the drug[30,58]. In addition, all liver function tests in experimental animals were found to be at wild-type levels at four and 5 months post-treatment, with AFP levels showing an age-appropriate decrease that was maintained throughout the study.

By the end of the study (337 days), liver histology in our experimentally treated animals mirrored the normal hepatic architecture found in our wild type age-matched controls. We found no evidence of chronic liver damage or increased AFP staining in our experimental animals even after 8 months off the protective drug NTBC. This is

especially important because we have also demonstrated here that $FAH^{-/-}$ animals that do not receive gene therapy and are chronically undertreated with NTBC develop extensive fibrosis within weeks and can progress to hepatic adenomas within 6 months of life and HCC prior to a year of life, which builds on previous findings of fibrosis in 100% of animals and adenoma development with milder NTBC undertreatment[59]. Importantly, HT1 pigs develop HCC in a fibrosis-dependent manner like humans and unlike mice[37,60,61], which makes the FAH-deficient pig an ideal preclinical model to approximate the potential deleterious effects of lentiviral therapy and NTBC treatment/withdrawal in humans. Previous studies using a non-lentiviral retrovirus (GlFSvNa) in mice to treat HT1 in vivo showed metabolic correction of the phenotype but increased rates of HCC. At that time the assumption was made that the increased HCC rates were due to a number of uncorrected hepatocytes that persisted and then progressed to HCC. This safety concern has precluded gene therapy development for HT1 for more than two decades, while our recent data in the more clinically relevant pig model show that this concern is outdated. The reasoning for this comes from our updated understanding of how HCC develops in mice, pigs, and humans.

Mice develop adenomas and HCC through a different set of molecular derangements when compared to humans and pigs; in fact,

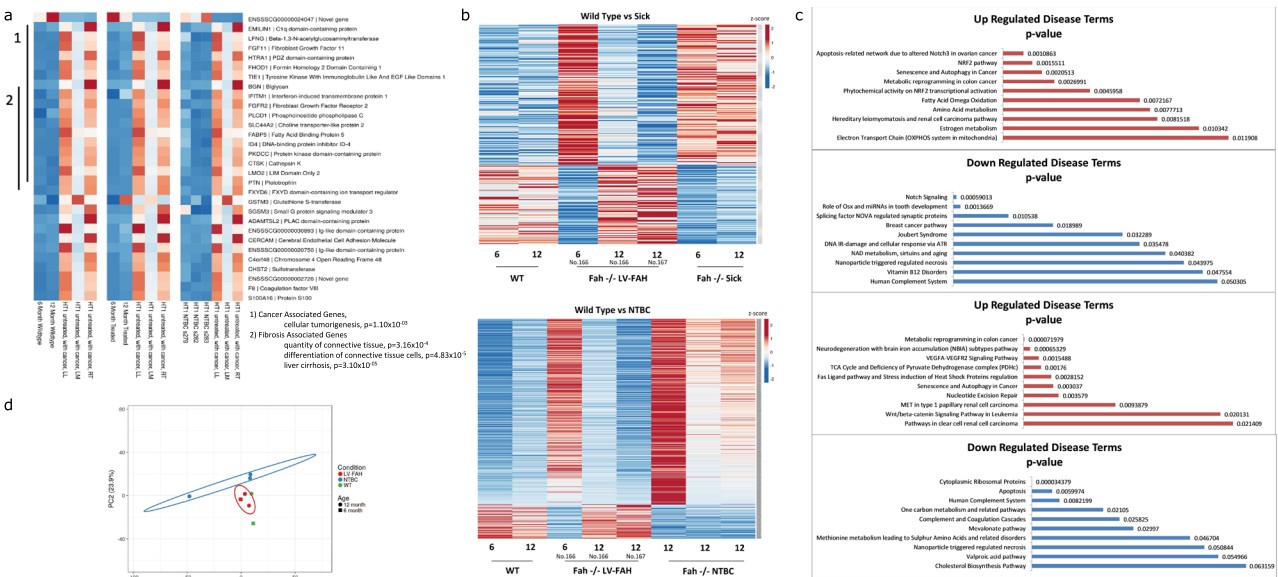

**Fig. 5 | RNA expression analysis shows LV-FAH restores wild type expression profiles better than standard NTBC treatment. a** Individual heatmaps comparing RNA expression levels of the genes listed on the right between wild-type pigs at 6 and 12 months of age (left), LV-FAH-treated pigs at 6 and 12 months of age (middle) and NTBC-treated pigs at 12 months of age (right). Genes displayed are the result of overlapping differential expression analyses across the three comparisons with absolute log2 fold change > 2 and FDR < 0.05. Clusters are identified for 1. cancer- and 2. fibrosis-associated genes. **b** Individual heatmaps showing RNA expression level profiles comparing 6 and 12-month wildtype and LV-FAH-treated pigs with either *FAH*[−/−] NTBC-undertreated pigs (Sick, top) or NTBC-treated pigs (bottom). **c** Gene clusters represented in (**b**) organized by association with various disease and cancer pathways showing that over the course of 12 months of LV-FAH

treatment genes in these pathways are expressed more similarly to wildtype than to the sick or standard NTBC-treated pigs. *P*-values were derived from an over-representation analysis with two-tailed tests without adjustment for multiple comparisons, and the genes shown in the figure the top 10 hits for each comparison from each direction. **d** Principal component analysis (PCA) supporting the conclusion that wild-type pigs are transcriptome-wide more similar to the LV-FAH treated pigs than to the NTBC-treated pigs. Original values are ln(x + 1) transformed. *X* and *Y*-axis show principal component 1 and principal component 2 that explain 46.4% and 23.9% of the total variance, respectively. Prediction ellipses are such that with probability 0.95, a new observation from the same group will fall inside the ellipse.

the pig model of HCC has been shown to develop HCC through a similar set of molecular derangements as seen in most human HCCs, including increased TP53 and KRAS expression, sustained angiogenesis resulting from overexpression of *PDGFA* and *ANGPT2*, evasion of apoptosis through overexpression of *TWIST1* and other mechanisms, reactivation of *TERT* for telomere maintenance, and Wnt signaling activation[62,63]. In humans, including HT1 patients, HCC develops in the setting of an extensively inflamed, fibrotic, or cirrhotic liver with the exception of the fibrolamellar variant or rare cases of HBV-related oncogenicity[64], while in *FAH*[−/−] mice HCC has been found to appear in the setting of no such alterations[32]. This is consistent with the observation that the tissue setting in which HCC develops is markedly different in humans and rodents, where chronic hepatitis and cirrhosis are uncommon lesions in the livers of rats and mice that develop HCC[36]. Therefore, mice develop adenomas and HCC in the liver through an inflammatory and fibrotic-independent process, where in humans and pigs, especially in the setting of HT1, adenomas and HCC develop almost exclusively in the backdrop of chronic fibrosis and cirrhosis, making the absence of these findings in the liver protective. We show histologically that by the end of therapy there is no evidence of any chronic inflammatory changes anywhere in the liver and this finding is supported by the RNA sequencing results discussed later. We also did not find any evidence of adenomas or HCC in either of our long-term experimental animals, consistent with our results with ex vivo therapy[30], nor did we find enrichment of any tumor-related pathways on GSEA. In addition, these animals were cycled on and completely off NTBC; this regimen could in theory be altered to involve chronic low doses of the drug, for example, in order to reduce the risk of fibrosis as FAH-positive hepatocytes expand to repopulate the liver, as well as avoid the porphyria-like crises and coagulopathy that untreated FAH deficiency causes in HT1 patients[65].

Further advocating for the safety of this approach, the LV-FAH vector showed a benign integration profile in pig hepatocytes after in vivo delivery. Evaluation of 17,000–39,000 unique integration sites in all animals revealed no genotoxic events and neither favored integration in tumor-coding regions of the genome or in CpG-rich promoter regions, and this correlation remained uniform for all liver tissue analyzed (Supplementary Table 3). This is consistent with previous studies, where lentivirus tends to integrate into gene coding regions without an obvious preference for transcription start sites, promoter regions or other regulatory sites of the genome[66,67]. These data, taken together with our previous work showing no evidence of genotoxicity of in vivo LV-FAH in an HCC-prone mouse model[68] and GSEA showing only enrichment of the PI3K-Akt and TGFb signaling pathways in one animal, establish a favorable safety profile, particularly since hepatocytes have been shown to be susceptible to insertional mutagenesis[69,70]. In our study, we observed integration bias in liver DNA beyond the well-established expectations reported previously[71]. We found that in addition to these previously reported biases, LV-FAH integration was increased at the *FAH* and *SERPINA1* loci, which may represent an integration bias based on sequence homology.

Gene expression associated with inflammation, fibrosis, and HCC in the liver has been well characterized[72]. RNA-seq analysis at 1 year of age from undertreated NTBC animals that had developed HCC was compared to age-matched controls of wild-type, adequately NTBC-maintained, and LV-FAH treated pigs. Relative to each other, there were significant patterns involving inflammation, fibrosis, and cancer-related genes that suggest LV-FAH treatment restores a wild-type phenotype by 12 months post-treatment. Remarkably, animals treated adequately with NTBC showed increases in these liver injury and tumor-associated genes, which may explain the robust data now available in humans demonstrating development of cirrhosis and HCC

in HT1 patients despite optimal and compliant NTBC therapy[7–10,73,74]. In addition, like wild-type animals, pigs treated with LV-FAH have normal, undetectable levels of AFP at 1 year, which is in stark contrast to undertreated NTBC FAH-deficient animals. This further supports our proposal of LV-FAH as an acceptable advancement in the treatment of HT1 in lieu of chronic NTBC maintenance therapy, which does not prevent development of inflammation, fibrosis, and HCC due to residual progression of tyrosine metabolism through the degradation pathway resulting in production of toxic metabolites. Furthermore, FAH-deficient patients optimally treated with NTBC are relegated at best to suffer from the HT3 phenotype, which can be debilitating over time[21–23,26]. HT1 patients are still in need of a true cure, such as that offered by LV-FAH.

Finally, the most compelling aspect of this model and disease is the characteristic expansion of FAH-positive hepatocytes. Genetically modified hepatocytes in the HT1 pig model provide an exquisitely sensitive tool to assess gene therapy platforms, including LV, for mutagenesis with subsequent robust clonal expansion of any deleterious clones. We provide evidence here that the HT1 pig model can develop HCC in the absence of gene therapy and adequate NTBC dosing, establishing the significance of the absence of tumorigenic integration events or activation of oncogenic pathways in LV-FAH treated animals and providing convincing safety data for consideration of human application. This work contributes to a growing body of evidence that LV vectors have a lower genotoxic risk when compared to gamma-retroviral vectors[75–77]. Although multiple strategies have and are being employed to increase the safety of lentiviral vectors in general[78,79], patients participating in clinical trials involving lentiviral vectors will continue to require close long-term surveillance. It is worth noting that HT1 patients already require frequent cancer-related surveillance and HCC screening[5], and this requirement would not be fundamentally altered after curative gene therapy to treat HT1.

There are a number of limitations to our study. The main and most significant limitation is the small number of animals treated and analyzed. As is often the case with preclinical large animal studies, inclusion of a large sample size would be cost-prohibitive. Results must be interpreted in the context of limited data obtained from a small number of animals followed for a relatively short period of time. However, the importance of testing new therapies in preclinical models that, unlike rodent models, fully recapitulate the human disease phenotype must not be undervalued, compounded with the fact that this particular model has demonstrated the ability to develop HCC within the relatively short timeframe studied. Doses were based on previous liver-directed in vivo LV large animal studies[48], as well as our own results in mice. Further dose-response studies are warranted to investigate the effects of dose on biodistribution and to find a balance between rapid cure with limited NTBC cycling and mitigation of systemic exposure leading to extrahepatic integration events. These further lowest-dose efficacy studies, which are outside the scope of this manuscript, are currently being performed. The doses used in this study would require large-scale manufacturing in adult or pediatric human patients, but new technology such as hollow fiber reactors and suspension culture processes are very rapidly decreasing manufacturing hurdles[80,81]. Strategies to decrease the LV dose needed could also include temporary Kupffer cell knockdown prior to LV administration in order to decrease vector clearance in the liver[82]. These further studies will be crucial to establish optimal in vivo lentiviral treatment protocols for clinical use. In addition, experimental pig livers did not undergo CT or MRI imaging, but rather gross histological examination after 0.5 cm sectioning. Therefore, it is possible that tumors under 0.5 cm in size were present, although absence of fibrosis and normal AFP levels as well as normal hepatic gene expression profile make this unlikely. Furthermore, while PCR as a tool to evaluate vector biodistribution can detect the presence of LV in a certain tissue, it does not differentiate between cell types within that tissue.

Therefore, when we assess biodistribution and integration profile within an organ, we are talking in broad terms about all the cells contained within that tissue. In the liver specifically, we are evaluating integration within hepatocytes, but also presumably stellate cells, Kupffer cells, and liver sinusoidal endothelial cells. Pigs were also not surveilled for HCC development across their entire lifespan. However, time to development of HCC in human patients off NTBC is certainly modeled, and the absence of fibrosis, elevated AFP, or concerning hepatic gene expression changes after over 6 months off NTBC are protective against future tumor development.

In conclusion, our study provides further evidence that in vivo lentiviral liver-targeted gene therapy via direct percutaneous portal vein delivery is safe and effective. These data support the consideration of in vivo administration of lentiviral vectors for the treatment of HT1 and other disorders that could be cured with stable integrating gene delivery.

## Methods

### Vector production

We have designed a LV vector containing the human *FAH* gene under control of a liver-specific promoter (HCR-AAT; hepatic control region enhancer and alpha-1 antitrypsin promoter), currently being trialed in humans for the treatment of hemophilia B[40]. A schematic representation of this vector is provided in Fig. 1a. In order to generate viral vectors, the LV-SFFV-eGFP or LV-AAT-huFAH expression construct, together with the packaging plasmid p8.91 and the vesicular stomatitis virus glycoprotein G-encoding plasmid pVSV-G, was transfected into 293 T/17 cells (CRL-11268, ATCC, Manassas, VA) using 1 mg/ml polyethylenimine (Polysciences, Warrington, PA). Viral supernatant was harvested 48 and 72 h after transfection, filtered through a 0.45-µm filter, and concentrated by ultracentrifugation (70,000 g, 1.5 h at 4 °C). After resuspension in serum-free media (DMEM, Thermo Fisher Scientific, Waltham, MA), LV vectors were aliquoted and stored at −80 °C. Vector titers were determined by p24 enzyme-linked immunosorbent assay and qPCR using the Lenti-X Provirus Quantitation Kit (Clontech, Mountain View, CA).

### Animals and animal care

For pig gene therapy experiments, male and female *FAH*−/− pigs were used. *FAH*−/− pigs were produced in a 50% Large White and 50% Landrace pig. *FAH*+/− pigs were produced through somatic cell nuclear transfer and were bred to produce *FAH*−/− pigs[59,83,84]. All pigs used in this and previous studies belong to the same original herd after some outbreeding. To obtain a homozygous age-matched cohort, heterozygous females were bred with homozygous males, yielding an expected prevalence of homozygosity in the litter of 50%. After birth, a notch of ear tissue from each piglet was obtained for genotypic confirmation through PCR as previously described[83,84]. Piglets determined to be homozygous were reserved for experimental purposes.

NTBC was obtained courtesy of Dr. Vadim Gurvich (University of Minnesota, Minneapolis, MN) and was administered mixed in food at a dose of 1 mg/kg/day with a maximum of 25 mg/day. All animals remained on NTBC until the time of LV-FAH treatment, after which NTBC administration was discontinued. Pigs were monitored daily for weight loss, and NTBC was reinitiated for 7 days if weight loss of 15% or more of body weight occurred or other signs of morbidity were present. Animals were cycled on and off NTBC in this fashion to stimulate expansion of corrected FAH-positive cells. Phenotype correction was assumed when an animal was able to thrive without NTBC treatment. For pig biodistribution experiments, male and female wild-type domestic pigs were used (Manthei Hog Farm, Elk River, MN).

Animals were housed inside Mayo Clinic's vivarium with standard light-dark cycles, temperature maintained between 61 and 81 degrees Fahrenheit, and humidity maintained between 30 and 70% depending on the season. Their welfare was monitored daily by Mayo Clinic

veterinary staff per internal institutional protocols. In addition, after receiving LV-FAH treatment, animals were weighed daily by a member of our team, checked for signs of liver failure or infection such as lethargy that would necessitate further veterinary care, and were administered NTBC if appropriate. Euthanasia was performed with intravenous pentobarbital injection following intramuscular sedation with Telazol at 5 mg/kg and Xylazine at 2 mg/kg per our Institutional Animal Care and Use Committee protocol. Live blood draws and other procedures were performed under sedation with the same agents, with 1–3% inhaled isoflurane being used for anesthetic maintenance during longer procedures.

## Pig experiments

Pigs were pre-treated according to a previously established immuno-suppression protocol[48]: 1 mg/kg oral prednisone, 1 mg/kg intramuscular diphenhydramine, and 0.5 mg/kg intramuscular famotidine the night before and the morning of the procedure, followed by 0.2 mg/kg IV dexamethasone and the same antihistamine regimen immediately prior to the procedure. Ear vein injection of LV-FAH was performed in one 6-week-old $FAH^{-/-}$ pig, at a dose of ~$2 \times 10^{10}$ transducing units (TU)/kg. In this case, the vector solution was infused through a 24-gauge intravenous catheter. As systemic LV-FAH administration led to a systemic inflammatory response syndrome (SIRS)-type response in this animal (see Results sections), ear vein injection of LV-GFP at the same dose was performed in one animal to ensure that the hypotension seen with systemic LV-FAH was a product of the administration method and not of the vector itself. Portal vein injection of LV-GFP at the same dose was performed in one heterozygous $FAH^{+/-}$ pig, also at 6 weeks of age.

Portal vein injections of LV-FAH at a dose of approximately $2 \times 10^{10}$ transducing units (TU)/kg were then performed in four $FAH^{-/-}$ pigs at 6 weeks of age. The portal vein was identified using a 2–5 MegaHz transducer (Fujifilm SonoSite, Inc., Bothell, WA), and an 18-gauge 5-in needle was directed percutaneously towards the main portal vein prior to its bifurcation for manual infusion of the vector solution.

Two reference pigs were evaluated at 1 year of age (Nos. 266 and 158) to demonstrate the development of HCC in this $FAH^{-/-}$ pig model. No. 158 was cycled on and off an appropriate dose of NTBC, while No. 266 was chronically underdosed to allow progression of the disease without mortality (Fig. 2a).

## Biochemical analysis

Complete blood counts were determined by analysis with the VetScan HM5 analyzer (Abaxis, Union City, CA) according to the manufacturer's instructions. Liver function tests and alpha-fetoprotein (AFP) levels were determined in serum, and ammonia levels were determined in plasma using standard protocols by the Mayo Clinic's central clinical laboratory. Tyrosine values in plasma were determined using liquid chromatography and tandem mass spectrometry by the Mayo Clinic's biochemical genetics laboratory. TNF-alpha, IL-6, and IL-8 levels were determined in serum through enzyme-linked immunosorbent assay by the Mayo Clinic's immunochemical laboratory. Liver function tests and tyrosine levels were compared to historical untreated $FAH^{-/-}$ ($n = 6$) and wild-type ($n = 2$) age-matched controls.

## Histopathological analysis

At the time of liver biopsy, two liver samples were collected, one from the left and one from the right lobes. After euthanasia, liver samples were collected from sixteen different areas spanning the entire organ. These areas were labeled as follows: left lateral lobe anterior superior, left lateral lobe anterior inferior, left lateral lobe posterior superior, left lateral lobe posterior inferior, left medial lobe anterior superior, left medial lobe anterior inferior, left medial lobe posterior superior, left medial lobe posterior inferior, right medial lobe anterior superior, right medial lobe anterior inferior, right medial lobe posterior superior, right medial lobe posterior inferior, right lateral lobe anterior

superior, right lateral lobe anterior inferior, right lateral lobe posterior superior, and right lateral lobe posterior inferior. For histological analysis, tissue samples were fixed in 10% neutral buffered formalin (Azer Scientific, Morgantown, PA) and processed for paraffin embedding and sectioning. H&E and Masson's Trichrome staining were prepared by means of standard protocols. H&E and Trichrome slides were evaluated by a pathologist for presence of liver injury and fibrosis. FAH immunohistochemistry using a polyclonal rabbit anti-FAH primary antibody[85] (FAHOR026; 1:2000 diluted in Bkg Reducing Diluent (S3022, Dako/Agilent, Santa Clara, CA, USA), incubated for 15 min) was performed with a Bond III automatic stainer with a 20-min antigen retrieval step using Bond Epitope Retrieval Solution 2 (Leica, Buffalo Grove, IL), and stained with diaminobenzidine (Leica, Buffalo Grove, IL). GFP immunohistochemistry using a monoclonal rabbit anti-GFP primary antibody (#2956, Cell Signaling, Danvers, MA; 1:100 diluted in Bkg Reducing Diluent, incubated for 15 min) and Ki-67 immunohistochemistry using a monoclonal anti-Ki67 primary antibody (MIB-1, M7240, Dako/Agilent, Santa Clara, CA, USA); 1:400 diluted in BOND primary antibody diluent (AR9352, Leica Biosystems Inc, Buffalo Grove, IL) were performed using the same process. TdT-mediated dUTP-biotin nick end labeling (TUNEL) and staining was also performed by the Mayo Clinic pathology research core with a TUNEL kit (Abcam, Cambridge, UK) according to the manufacturer's instructions. H&E and Trichrome slides from each of the sixteen liver areas sampled were reviewed by an independent blinded liver pathologist that estimated the percent of fibrotic liver tissue in each slide examined. Fibrosis was quantified using the previously described porcine METAVIR scoring system[55]. Quantification of FAH-positive cells was performed using cytoplasmic stain algorithms in Aperio ImageScope. Entire slides from each lobe of the liver were analyzed and quantified for each animal. Reported results are the total percentage of cytoplasmic FAH positivity among the cells in each slide. Quantification of Ki-67 and TUNEL-positive cells were hand counted: for each sample, 10 high powered fields at 40x magnification were counted. All fields were 100 µm × 100 µm with positive and total cells hand counted. Only hepatocytes were counted; stromal cells as well as biliary epithelial cells were excluded.

## Vector integration analysis

Genomic DNA was isolated from snap-frozen tissue fragments using a Gentra Puregene Tissue Kit (Qiagen, Hilden, Germany). Representative samples were taken from sixteen distinct areas of the liver and pooled for analysis. Three pairs of primers were designed to amplify integrated vectors in the pig genome, targeting the 3′ LTR and 5′ LTR regions as well as the transgene of interest. For LV-GFP, these primers were: 3′ LTR (5′-CTGTTGGGCACTGACAATTC-3′ and 5′-TAACTAGAGATCCCTCAGACCC-3′), 5′ LTR (5′-GGATGTCTTCAATCAGCCTA-3′ and 5′-GGCTTAGAGTCATCAGGTTT-3′), and GFP (5′-CCTGAAGTTCATCTGCACCA-3′ and 5′-GACAACCACTACCTGAGCAC-3′). For LV-FAH, these primers were: 3′ LTR (5′-TGTGACTCTGGTAACTAGAGATCCCTC-3′ and 5′-TTGCCTTGGTGGGTGCTACTCCTAATG-3′), 5′ LTR (5′-GGATGTCTTCAATCAGCCTA-3′ and 5′-GGCTTAGAGTCATCAGGTTT-3′), and FAH (5′-TGTTGGAACTGTCGTGGAAG-3′ and 5′-AGCAGTGGGTTCCCTAGTTA-3′). PCR was performed with GoTaq DNA Polymerase (Promega, Madison, WI) using an initial denaturation step at 98 °C for 3 min, followed by 30 cycles at 98 °C for 30 s, 58 °C for 30 s, and 72 °C for 40 s, and finally at 72 °C for 5 min. PCR products were visualized by means of 2% TBE agarose gel electrophoresis with ethidium bromide staining.

## Amplification, next-generation sequencing (NGS), and genomic DNA mapping of lentiviral integration sites

Genomic DNA was isolated from snap-frozen tissue fragments using a Gentra Puregene Tissue Kit (Qiagen, Hilden, Germany). Ligation-mediated PCR (LM-PCR) was used for efficient isolation of integration

sites. Restriction enzyme digestions with MseI were performed on genomic DNA samples; the digested DNA samples were then ligated to linkers and treated with ApoI to limit amplification of the internal vector fragment downstream of the 5′ LTR. Samples were amplified by nested PCR and sequenced using the Illumina HiSeq 2500 Next-Generation Sequencing System (San Diego, CA). After PCR amplification, amplified DNA fragments included a viral, a pig and sometimes a linker segment. The presence of the viral segment was used to identify reads that report a viral mediated integration event in the genome. The reads sequenced from these DNA fragments were processed through quality control, trimming, alignment, integration analysis, and annotation steps.

Quality control of the sequenced read pairs was performed using the FASTQC software. The average base quality of the sequenced reads was >Q30 on the Phred scale. Uniform distribution of A,G,T and C nucleotides was seen across the length of the reads without bias for any specific bases. The number of unknown "N" bases was less than 1% across the length of the reads. High sequence duplication (>80%) was observed, however, this was anticipated due to the nature of the experiment and the amplification of specific library fragments.

We used Picard software's (http://broadinstitute.github.io/picard/) insert size metrics function to calculate the average fragment length per sample. This metric averaged across all the samples was 158 base pairs (bps) long with an average standard deviation of 29 bps. Since 150 bp long reads were sequenced, most of the paired read one (R1) and read two (R2) reads overlapped, providing redundant information. R2 reads were therefore not considered in the analysis.

Reads were trimmed to remove the viral sequence in two separate steps. In step 1, the viral sequence was trimmed from the R1 reads using cutadapt[86] with a mismatch rate ($e = 0.3$) from the 5′ end of each read. In step 2, if the linker sequence was present, it was similarly trimmed from the 3′ end of the read. Trimmed reads with a length less than 15 bps were removed from the rest of the analysis to reduce ambiguous alignments. Untrimmed reads were also removed from the rest of the analysis because they did not contain a viral segment that could be used as evidence of a viral mediated integration.

The remaining R1 reads were aligned to the susScr11 build of the pig reference genome using BWA-MEM in single-end mode[87]. Default BWA-MEM parameters were used. The reads used to identify genomic points of viral integrations had to be uniquely mapped to the genome with a BWA-MEM mapping quality score greater than zero.

An integration point was defined by the position of R1's first aligned base on the susScr1 genome. Unique integration points were identified across the genome without any constraints on coverage. However, for downstream annotation and analysis, only those integration points with 5 or more supporting reads were used to minimize calling false positive integration points.

Locations of integration points were categorized into: exons, introns, 3 prime UTRs, 5 prime UTRs and intergenic regions using information extracted from the susScr11 refflat file maintained by UCSC's Genomics Institute. To avoid conflict of feature categorization arising from multiple overlapping gene definitions, only the definition of the longest gene was used to annotate integration points. Additional features where computed including: the distance to the nearest transcription start site (TSS), makeup of CpG-rich regions of the genome ("CpG islands"), and enrichment to tumor-associated genes. This tumor gene list was comprised of 745 clinically validated tumor-associated human genes (from Mayo Clinic internal data). Those genes were related to their pig homologs based on GenBank's gene names.

In order to look for selective enrichment of any specific biological pathways, the genes identified as integration sites were entered into the WEB-based GEne SeT AnaLysis Toolkit[88]. Results were identified using the over-representation analysis (ORA) method with identified KEGG pathways. A false discovery rate (FDR) of <0.05 was considered significant.

## RNA sequencing

Liver samples for RNA sequencing were obtained at necropsy from multiple different areas spanning the entire liver, as described in the Histopathological Analysis section. Samples were taken from two wild-type animals and three $FAH^{-/-}$ animals adequately treated with NTBC, as well as our two long-term FAH-treated experimental animals and chronically undertreated pig 266. These samples were flash frozen and stored at −80 °C.

Samples were provided to the Mayo Clinic Sequencing Core, where total RNA was extracted using RNeasy Midi extraction kit (Qiagen) by following the manufacturer's instructions. After RNA sample quality assessment, poly-A selection and fragmentation of 100 nanograms to 1 microgram of starting total RNA was performed. The fragments then underwent reverse transcription with random primers and second-strand synthesis to generate double-stranded cDNA. Ends were repaired and adenylated, followed by adapter and index ligation. Products were denatured and PCR-enriched to generate the final genomic library, which underwent quality control for quantitative and qualitative thresholds prior to sequencing on the Illumina HiSeq 2000 Next-Generation Sequencing System (San Diego, CA) generating paired end reads.

RNA sequencing data was processed through the Mayo Analysis Pipeline for RNA sequencing (MAP-Rseq), which provides gene counts, exon counts, fusion candidates, expressed single nucleotide variants, mapping statistics, visualizations, and a detailed research data report for RNA-Seq[89]. Within MAP-RSeq, the mRNA-Seq data were aligned with STAR version 2.5.2b[90] to the Sscrofa11.1 assembly of the pig genome. The raw gene expression within MAP-RSeq was then quantified with the featureCounts software[91]. Normalization and differential expression analyses were performed using R version 3.5.3 with the edgeR version 3.24.3[92] and limma version 3.38.3 packages. To identify differentially expressed genes, median, minimum, and maximum values were extracted from the expression data across the samples (total of 25,149 genes), removing all genes with median expression less than 25 across all samples and minimum expression of 0 (remaining total of 4288 genes), and calculating a fold change comparing minimum and maximum expression and removing all genes with a fold change less than 10. This strategy was used to ensure the genes evaluated for biological significance were reliably expressed across all samples, and that the levels of expression were variable. The remaining genes ($n = 30$) were then extracted based upon their pattern of normalized expression values trending towards cancer. Comparisons were performed between the following groups: (1) wild-type ($n = 2$), (2) $FAH^{-/-}$ pigs adequately maintained on NTBC ($n = 3$), (3) $FAH^{-/-}$ sick pig No. 266, and (4) experimentally treated pigs Nos. 166 and 167. Three-way differential expression analyses between these groups were performed, and genes displayed in the manuscript figures are the result of overlapping differential expression analyses across the three comparisons with absolute log2 fold change > 2 and FDR < 0.05. Network and functional term enrichment analyses were processed through Qiagen's Ingenuity application[93] on the identified differentially expressed genes. Heat-maps visualizing the z-scores of the relevant genes' normalized expression values were created using the clustvis web tool[94]. Principal component analysis (PCA) was also performed with Nipals PCA used to calculate principal components and Pareto scaling applied to rows.

## Statistics

Numerical data are expressed as individual data points. Biochemical and histological comparisons between groups were performed using Welch's t-test. $P < 0.05$ was considered statistically significant. Statistical analysis was performed using GraphPad Prism software version 7.03. Integration comparisons between groups were performed using a z-test and gene expression comparisons between groups were performed using an over-representation analysis.

## Study approval

All animal procedures were reviewed and approved by Mayo Clinic's Institutional Animal Care and Use Committee, and all animals received humane care for the duration of the study.

## Reporting summary

Further information on research design is available in the Nature Research Reporting Summary linked to this article.

## Data availability

The weight, biochemical, and numerical histological data generated in this study are provided in the Source Data file. The DNA sequencing and RNA-seq data generated in this study have been deposited in the GEO database under accession code GSE189135. The Sscrofa11.1 assembly of the pig genome by the Swine Genome Sequencing Consortium can be found on RefSeq under accession code GCF_000003025.6 or GenBank under accession code GCA_000003025.6. Source data are provided with this paper.

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

## Acknowledgements

Funding for this study came from Children's Hospitals of Minnesota and additional support was provided by Regenerative Medicine Minnesota. We thank L. Gross and L. Acosta for histology support; S. Krage and J Pederson for surgical support; G. Mondal for assistance with bioinformatics data.

## Author contributions

C.T.N. designed the study, conducted experiments, performed animal care, acquired and analyzed data, and wrote the manuscript. C.J.VL. conducted experiments and acquired and analyzed data. R.D.H., Z.D., R.M.G., W.J.C, B.H., A.L., and D.R. conducted experiments and acquired data. L.G.H. performed data collection and animal curation. A.B., D.O., and J.-P.K. analyzed data. S.J.R. analyzed data and wrote the manuscript. R.A.K. and J.B.L. designed the study, conducted experiments, analyzed data, and wrote the manuscript.

## Competing interests

J.B.L. is Chief Scientific Officer and R.A.K. is Vice-President of Pre-Clinical Development of Castle Creek Biosciences, Inc. as of January 10, 2022. No funding or financial support for this work has been provided by Castle Creek Biosciences. The remaining authors have no competing interests.
