## [Peer Review File · Nature Communications]

Reviewers' Comments:

Reviewer #1:

Remarks to the Author:

General comments

Nikolas et al. present data on safety and efficacy of lentivirus mediated expression of Fah in a pig model of tyrosinemia type I. The manuscript is well written and provides well-structured experiments and important data on this topic. Overall, they show superior safety of intraportal infusion of lentiviral vectors over peripheral vein injection and demonstrate efficacy of repopulation of the liver by gene corrected hepatocytes in two animals. They conclude from their experiments that intraportal application of lentiviral vectors is therapeutically superior to existing therapies, e.g. NTBC treatment, and support the view of superior safety of this approach.

Major comments

1. In general, the reviewer agrees with the conclusion that lentiviral gene transfer in pigs provides efficient repopulation of the liver by the gene corrected hepatocytes. Of more concern is the observed toxicity seen after peripheral vein injection, but also the elevated cytokine levels after intraportal application of the lentiviral vector. With the virus concentration used in their study, a narrow therapeutic window would exist, which limits the usefulness in the clinic. It is not clear from the manuscript, how the applied virus concentration, the volume, the period of injection, was determined. It would be mandatory to perform a dose finding study and define the lowest concentration of the virus to increase overall safety. Overall, current safety concern would not allow proposing a clinical study based on the parameters described in the manuscript.
2. The observation time of two animals was extended to approximately one year after virus injection. Although the authors see a near complete repopulation of the liver by gene corrected cells, it is difficult to draw the conclusion that HCC could be prevented. Given the lifespan of landrace pigs of more than 20 years prevention of HCC can only be assumed. However, normal gross pathology, absence of inflammation and infiltrates may point to a favorable outcome. A more balanced discussion would certainly increase the impact of the study. Additional biomarkers such as TUNEL or Ki67 staining showing absence of cell death and increased proliferation, respectively, would help to support the view that HCC or other tumor entities could be prevented.

Minor comments

1. Figure 1 and supplemental figure 1a are not well organized. In the main text an animal is mentioned, which was injected with LV.GFP systemically, but data are not shown. Please add the corresponding data (IL-6, TNFalpha etc.) either in Fig 1 or in supplement. Also show the blood pressure curve for the two animals with systemic administration.
2. In Fig. 2 the legend states that p-values for the biochemical markers are given for treated animals compared to Fah(-/-) animals, however two p-values are provided in the figures.
3. On page 10, first row the repopulation rate in the long-term animals was estimated as 69% and 78% of cells in the liver. Is it 69% and 78% of hepatocytes or total number of cells ?
4. Although the immunosuppression protocol has been cited exact timing and dosing should be mentioned in the text.

Reviewer #2:

Remarks to the Author:

Clara et al. performed human Fah delivered by lentivirus vector into the liver to cure hereditary tyrosinemia type 1 (HT1) in pigs and claimed this would prevent liver cancer induced by NTBC treatment. Previously, they applied the lentivirus into isolated hepatocytes and fixed the HT1 by corrected FAH-positive hepatocytes. In principle, in vivo delivery of lent-virus will be significant progress compared with the previous treatment. However, safety is a big issue, and the concerns for this manuscript are shown in the following.

0.1% hepatocytes expressing Fah will cure the HT1, thus to cure the HT1 disease is not a surprise. Compared with the corrected hepatocytes transplanted into the liver or AAV as a vector carrying Fah into the liver, the safety of the in vivo lentivirus delivery is a super concern. Besides the hepatocytes, the lentivirus will affect other cell types in the liver even if we assume all viruses come to liver tissue only. The authors should identify the off-targets by cell types.

Off-targets induced by lentivirus need long-term detection and more animals. These authors just applied one or two animals to support the conclusion. It is hard to convince me the detected off-

targets shown in the table are safe. Even for these two animals, the off-targets are diverse with time (shown in the figure, day 66 and day 337). These results suggest that the off-targets are out of control, and especially some off-targets locate in the exon regions. Compared with AAV, lentivirus will insert into the genome, and the authors detected around ten copy number of Lenti in each cell; they will affect another gene with time.

For only one or two animals, the authors could not get a statistical significance by using mean \pm SD. It is better to show each data point for every treatment and increase the animal number and time points

Why will the Lenti-FAH induce the benign in the liver after correcting the hepatocytes?

Reviewer #3:

Remarks to the Author:

This study seeks to confirm the safety and efficacy of in vivo lentiviral vector based delivery of the FAH gene in a transgenic porcine model of HT1. The pig represents an ideal model for studying this type of therapy given the similar size, metabolism, genetics, and immune responses between pigs and humans. In addition, the porcine HT1 model is more clinically relevant than the mouse model. This study is therefore of significant interest to the HT1 and gene therapy communities. However, there are several limitations, missing details, and analyses that need to be included in order to allow for full interpretation of the results. My specific comments are provided below.

- Line 186-187. Were histology samples scored using the METAVIR system? If so, what was the stage? If not this should be included.

- Line 196. A porcine METAVIR scoring system based on the human scoring system has been created. Would recommend using the porcine scoring system to assess fibrosis levels in these samples.

Gaba et al. Characterization of an inducible alcoholic liver fibrosis model for hepatocellular carcinoma investigation in a transgenic porcine tumorigenic platform (2018). *JVIR* 29(8):1194-1202

- Lines 199-205. METAVIR scores for all sections would provide a more quantitative assessment of fibrosis level. It is difficult to interpret differences in fibrosis level when the 60 day timepoint is only descriptive (no active fibrosis), the 225 day timepoint is listed as METAVIR Stage 1-2, and the 337 day timepoint is listed as minimal fibrosis in 10-15% of the liver.

- Lines 205-207. METAVIR scores for the adult wild-type pigs (and inclusion of number of controls used) should be provided for result interpretation.

- Line 208. Only a single pig is listed. Was this pig untreated or chronically undertreated? Was only 1 un/undertreated pig used as a comparison? How consistent are these results (METAVIR stage 4 with HCC tumor development within 12 months) in this model? How many HCC tumors developed, and what was their size? Are CT or ultrasound images available to confirm?

- Lines 211-213. It is difficult to effectively screen a 1 year old pig liver at necropsy via sectioning. Was CT or other imaging performed to confirm the lack of HCC tumor development in any of these pigs?

- Lines 320-323. A reference should be provided for the statement that untreated FAH $^{-/-}$ pigs develop fibrosis within weeks and progress to hepatic adenomas within 6 months and HCC within 1 year.

- Line 405. The lack of CT/MRI imaging to extensively profile for HCC or adenoma formation in these pigs should be listed as a limitation of the study. It should also be noted that longer term studies may be required to confirm HCC development has been eliminated as opposed to simply delayed.

- Given the limited number of animals studied (n=2 for 1 year), more details regarding the % of untreated/undertreated FAH $^{-/-}$ pigs that develop cirrhosis, adenomas, and HCC within 1 year should be provided. If 100% of untreated/undertreated FAH $^{-/-}$ pigs develop cirrhosis or HCC within 1 year, this is clearly a significant result. But if only 20% do, then the lack of cirrhosis or HCC development in the 2 animals studied could be due to chance.

- Line 443. How were FAH $^{-/-}$ pigs produced? Is this the same herd used in previous studies where cirrhosis and HCC tumor development has been demonstrated previously? If so a reference to the

paper describing their initial production should be provided.

- Line 468. Data from pig 158 is not described in the results. Did this pig also develop cirrhosis and HCC tumors within 1 year?

- Line 479-480. How many untreated FAH^{-/-} and wild type controls were used for this comparison? Were they age matched?

- Line 571. It's unclear how many/which samples from which animals were sequenced, and how these samples were collected and stored. How RNA was isolated, library preparation, sequencing information (platform and single vs paired end) is also not provided. Comparisons performed and number of differentially expressed genes for each comparison should also be provided.

- Figure 4 and 5 are hard to read due to the very small text.

- Figure 5a. Unclear why these genes were chosen to be displayed in this heatmap. Some are cancer or fibrosis-related, but not all and it's unclear the significance of the ones chosen.

- Figure 5b. Are these heatmaps based on all expressed genes or DEGs? Some sort of cluster analysis (heatmap, PCA) for all expressed genes would be valuable to confirm global expression is more similar between LV-FAH treated and WT compared to NTBC/untreated pigs.

- Why was RNAseq only performed on 1 of the pigs treated with LV-FAH and maintained for 12 months?

- Accession numbers for the genomic/RNA-seq data produced do not appear to be provided.

Reviewer #1 (Remarks to the Author):

General comments

Nikolas et al. present data on safety and efficacy of lentivirus mediated expression of Fah in a pig model of tyrosinemia type I. The manuscript is well written and provides well-structured experiments and important data on this topic. Overall, they show superior safety of intraportal infusion of lentiviral vectors over peripheral vein injection and demonstrate efficacy of repopulation of the liver by gene corrected hepatocytes in two animals. They conclude from their experiments that intraportal application of lentiviral vectors is therapeutically superior to existing therapies, e.g. NTBC treatment, and support the view of superior safety of this approach.

Major comments

1. In general, the reviewer agrees with the conclusion that lentiviral gene transfer in pigs provides efficient repopulation of the liver by the gene corrected hepatocytes. Of more concern is the observed toxicity seen after peripheral vein injection, but also the elevated cytokine levels after intraportal application of the lentiviral vector. With the virus concentration used in their study, a narrow therapeutic window would exist, which limits the usefulness in the clinic. It is not clear from the manuscript, how the applied virus concentration, the volume, the period of injection, was determined. It would be mandatory to perform a dose finding study and define the lowest concentration of the virus to increase overall safety. Overall, current safety concern would not allow proposing a clinical study based on the parameters described in the manuscript.

Response: Thank you for bringing this topic to light. As you know well, the FDA has described several parameters based on safety and toxicology related to *in vivo* gene therapy using viral carriers. Most of this information is related to AAV as LV's use to date has largely been *ex vivo*. We have worked with the FDA to design the experiments in this paper as part of the IND-enabling studies to clinic. Indeed, your point is very well taken that this paper and series of experiments were not meant to address dosing or all of the clinical aspects of a clinical protocol. This study was designed to test the issue of safety, specifically with regards to genotoxicity, which is a major area of concern for the FDA and a significant area of deficit in the literature with the *in vivo* use of lenti. This study demonstrates the major conclusions that portal vein delivery limits biodestruction and therefore off-target VCN remains exceedingly low. We also noted no SIRS response with portal vein delivery when compared to systemic vein delivery. More importantly, for this study using this construct we wanted to determine if a dominant clone would be observed after repopulation, as well as what the general molecular behavior of the liver was after complete repopulation with FAH+ transgenic cells. We showed in this unique large animal model that the liver did not have a dominant clone after repopulation and that the molecular character of the liver was essentially identical to an age-matched wild type control. This was a major hurdle for the FDA and represents novel data for the community to consider when thinking about the future of LV-directed gene therapy. Moreover, this study was designed specifically to use a dose that would be an order of magnitude or higher to prove this critical safety point and the viral dose delivered was determined based on previous *in vivo* LV liver-directed gene therapy studies such as the work by Cantore *et al* (Sci Trans Med, 2015). This has been clarified in the text (lines 156, 458). Our last IND-enabling studies are focused on dose as our pre-IND feedback believes we have addressed the major concern of genotoxicity. We wholeheartedly agree with the reviewer in that further dose safety studies would be required prior to clinical application to determine the lowest effective dose – those studies are being performed under guidance obtained at our pre-IND meeting with the FDA. In conclusion, the reviewer makes an important point in that the current work supports use of LV-FAH in HT1 patients, but it by no means a definitive IND-enabling body of work. Further preclinical development is indeed underway, but that work is outside the scope of this manuscript and our laboratory's ability to report on those findings. Nevertheless, we have included this in our discussion (line 462).

2. The observation time of two animals was extended to approximately one year after virus injection. Although the authors see a near complete repopulation of the liver by gene corrected cells, it is difficult to draw the conclusion that HCC could be prevented. Given the lifespan of landrace pigs of more than 20 years prevention of HCC can only be assumed. However, normal gross pathology, absence of inflammation and infiltrates may point to a favorable outcome. A more balanced discussion would certainly increase the impact of the study. Additional biomarkers such as TUNEL or Ki67 staining showing absence of cell death and increased proliferation, respectively, would help to support the view that HCC or other tumor entities could be prevented.

Response: We appreciate the reviewer's feedback and agree these additional studies would enhance the data underlying the conclusions we make in the manuscripts. We have performed and added TUNEL and Ki67 staining to liver sections from treated animals, as well as wild-type and undertreated *FAH*^{-/-} controls (Suppl Fig 6), and believe this has increased the strength of the evidence to support the idea that HCC

prevention in HT1 can be achieved through gene therapy. Although as the reviewer points out we are not able to follow our experimental animals for their entire lifespan, the liver damage that results from NTBC withdrawal in FAH^{-/-} pigs leads to adenoma and HCC formation by a year of life (as exemplified by pig 266), which is how long we followed our *in vivo*-treated experimental animals for. In clinical practice, adenomas and/or HCC occur in human patients within this timespan when not treated with NTBC. We have been working with these breeds of pig for over a decade and have not been able to keep wild-type or fully treated animals alive beyond three years in captivity/lab (hard surface and grated surface living space) as these animals essentially gain weight continuously throughout their life leading to morbidity related to hind limb and hoof issues and skin break down. These points have been added to the discussion of study limitations (line 472).

Minor comments

1. Figure 1 and supplemental figure 1a are not well organized. In the main text an animal is mentioned, which was injected with LV.GFP systemically, but data are not shown. Please add the corresponding data (II-6, TNFalpha etc.) either in Fig 1 or in supplement. Also show the blood pressure curve for the two animals with systemic administration.

Response: We appreciate the reviewer's attention to these unrepresented data. We have added details about the LV-GFP systemic pig and rationale for injection to our methods section (line 529) and have included both systemic animals' BP curves as well as heart rate curves in Suppl Figure 1. These were not included in the original manuscript because unlike animals who received portal vein delivery, systemic administration required the short term use of pressors which artificially elevated/maintained BP in the animals. Per FDA request, the systemic LV-GFP animal was injected as a way to ensure that the hypotension seen with systemic LV-FAH was a product of the administration method and not of the vector itself. Therefore, this animal was unfortunately not maintained under general anesthesia post-injection in order to obtain inflammatory markers.

2. In Fig. 2 the legend states that p-values for the biochemical markers are given for treated animals compared to Fah^(-/-) animals, however two p-values are provided in the figures.

Response: We appreciate the reviewer's attention to detail. The legend for Fig 2 has been corrected.

3. On page 10, first row the repopulation rate in the long-term animals was estimated as 69% and 78% of cells in the liver. Is it 69% and 78% of hepatocytes or total number of cells ?

Response: These numbers represent total number of cells. This has been clarified in the text.

4. Although the immunosuppression protocol has been cited exact timing and dosing should be mentioned in the text.

Response: Thank you for this feedback. The exact timing and dosing of the immunosuppression protocol have been added to the methods section (line 524).

Reviewer #2 (Remarks to the Author):

Clara et al. performed human Fah delivered by lentivirus vector into the liver to cure hereditary tyrosinemia type 1 (HT1) in pigs and claimed this would prevent liver cancer induced by NTBC treatment. Previously, they applied the lentivirus into isolated hepatocytes and fixed the HT1 by corrected FAH-positive hepatocytes. In principle, *in vivo* delivery of lent-virus will be significant progress compared with the previous treatment. However, safety is a big issue, and the concerns for this manuscript are shown in the following.

0.1% hepatocytes expressing Fah will cure the HT1, thus to cure the HT1 disease is not a surprise. Compared with the corrected hepatocytes transplanted into the liver or AAV as a vector carrying Fah into the liver, the safety of the *in vivo* lentivirus delivery is a super concern. Besides the hepatocytes, the lentivirus will affect other cell types in the liver even if we assume all viruses come to liver tissue only. The authors should identify the off-targets by cell types. Off-targets induced by lentivirus need long-term detection and more animals. These authors just applied one or two animals to support the conclusion. It is hard to convince me the detected off-targets shown in the table are safe. Even for these two animals, the off-targets are diverse with time (shown in the figure, day 66 and day 337). These results suggest that the off-targets are out of control, and especially some off-targets locate in the exon regions. Compared with AAV, lentivirus will insert into the genome, and the authors detected around ten copy number of Lenti in each cell; they will affect another gene with time.

For only one or two animals, the authors could not get a statistical significance by using mean \pm SD. It is better to show each data point for every treatment and increase the animal number and time points.

Why will the Lenti-FAH induce the benign in the liver after correcting the hepatocytes?

Response: Thank you for this feedback and thorough review. We agree fully with the reviewer that HT1 is quite unique in that a very small number of corrected cells will be able to repopulate the liver. In fact, this is one of the most interesting aspects to this study and likely one of its most significant conclusions. In typical viral direct gene therapy approaches, targeted and off-targeted cells will divide a limited number of times (if at all for some cell types) leading to lingering concerns years down the road about genotoxic events. In HT1 we see the product of hundreds and thousands of cell divisions which would test and strain any gene therapy approach or platform to reveal potential genotoxic events at much earlier timepoints. This is quite powerful! It is worth noting that we did not see the development of a dominant clone, even at the high dose used in this study. We have worked extensively with the FDA to design these studies and this result in our large animal models has made it possible to proceed on to FIH studies. In fact, the last remaining IND enabling studies are related to finding the lowest safe and effective dose to allow for a clinical program that will use LV *in vivo* to treat HT1 in humans.

We did not isolate different liver tissue cell types when performing this analysis. However, we did look at whole liver tissue including all the stromal cells. In that analysis, we did not see a shift in molecular behavior of the whole organ (Fig 5). We further looked at other off-targets through PCR and found no integration into these tissues. We performed off-target analysis in four different large animals, confirmed with 3 different primer pairs. The first animal was observed at 48 hours, the second at 60 days, and the last 2 at nearly one year after therapy. We developed this approach while working with the FDA on early and late time points for the detection of off-target integration events. We can see essentially in four large animals there are no significant off-target effects. We compare this to systemic administration which shows a much different profile, where LV targets the liver poorly and targets many other organ systems. We wholeheartedly agree that more animals and increasing the number of observations would increase our comfort with any conclusion, but for the scope of this study we limited our approach to looking at off-target activity of LV after portal vein administration to four large animals. We believe this addresses the reviewer's concerns not just in the liver but in other potential organs, taking very seriously the thoughts this reviewer had for our manuscript. As touched upon in our discussion, inclusion of a larger amount of animals in this study to be followed for a year or longer would have been cost-prohibitive and not something the FDA required so long as no dominant clones were found. However, prior to clinical application of this technology, a minimal dose-efficacy study will be completed in accordance with FDA guidance. This dose-safety study is outside the scope of this project as it will focus on effectiveness and shorter-term safety of different doses. As you know well, the FDA has described several parameters based on safety and toxicology related to *in vivo* gene therapy using viral carriers. Most of this information is related to AAV as LV's use to date has largely been *ex vivo*. We have worked with the FDA to design the experiments in this paper as part of the IND-enabling studies to clinic. This study was designed to test the issue of safety, specifically with regards to genotoxicity, which is a major area of concern for the FDA and a significant area of deficit in the literature with the *in vivo* use of lenti. This study demonstrates the major conclusions that portal vein delivery limits biodistribution and therefore off-target VCN remains exceedingly low. More importantly, for this study using this construct we wanted to determine if a dominant clone would be observed after repopulation, as well as what the general molecular behavior of the liver was after complete repopulation with FAH+ transgenic cells. We showed in

this unique large animal model that the liver did not have a dominant clone after repopulation and that the molecular character of the liver was essentially identical to an age-matched wild type control. This was a major hurdle for the FDA and represents novel data for the community to consider when thinking about the future of LV-directed gene therapy. Moreover, this study was designed specifically to use a dose that would be an order of magnitude or higher to prove this critical safety point. Our work and others indicate that LV integration is benign, unlike gamma-retrovirus precursors. Although unlikely, it is theoretically possible to have a tumor-promoting outcome. We and others have tested this negative to a reasonable extent, but definitive data will only be available based on long term outcomes applicable only to clinical application. The value of the experimental models has been pushed to their useful limit with only supportive safety and efficacy data.

In summary, we agree with many of the reviewer's thoughtful comments. However, we believe that our data strongly suggests long-term safety in view of the following:

- 1) FAH-/- pigs develop adenomas and even HCC within a year of life if chronically undertreated with NTBC; our treated experimental animals remained off NTBC for 8-9 months prior to histological evaluation and gene expression profiling, with no tumorigenesis
- 2) FAH-/- pigs develop HCC in the context of fibrosis and elevated AFP, neither of which are present in our long-term experimental animals, predicting the durability of this finding beyond the experimental window
- 3) The LV-FAH integration profile within the liver was found to be comparable to that of previously published studies; to date no clinical trial involving ex vivo use of LV vectors in human patients has reported an integration-derived malignant adverse event.

We have altered our data presentation to show each individual data point for our experimental animal's biochemical data (Fig 2). NGS and RNA seq data is already presented individually by animal.

Reviewer #3 (Remarks to the Author):

This study seeks to confirm the safety and efficacy of in vivo lentiviral vector based delivery of the FAH gene in a transgenic porcine model of HT1. The pig represents an ideal model for studying this type of therapy given the similar size, metabolism, genetics, and immune responses between pigs and humans. In addition, the porcine HT1 model is more clinically relevant than the mouse model. This study is therefore of significant interest to the HT1 and gene therapy communities. However, there are several limitations, missing details, and analyses that need to be included in order to allow for full interpretation of the results. My specific comments are provided below.

- Line 186-187. Were histology samples scored using the METAVIR system? If so, what was the stage? If not this should be included.

Response: We appreciate the reviewer's feedback and acknowledge this is an important point. We have now included porcine METAVIR scores for all histology sections (Suppl Table 2).

- Line 196. A porcine METAVIR scoring system based on the human scoring system has been created. Would recommend using the porcine scoring system to assess fibrosis levels in these samples. Gaba et al. Characterization of an inducible alcoholic liver fibrosis model for hepatocellular carcinoma investigation in a transgenic porcine tumorigenic platform (2018). *JVIR* 29(8):1194-1202

Response: We have included porcine METAVIR scores for all histology sections and appreciate this reviewer's insight. The porcine METAVIR scoring system is now referenced in our methods (Ref 55).

- Lines 199-205. METAVIR scores for all sections would provide a more quantitative assessment of fibrosis level. It is difficult to interpret differences in fibrosis level when the 60 day timepoint is only descriptive (no active fibrosis), the 225 day timepoint is listed as METAVIR Stage 1-2, and the 337 day timepoint is listed as minimal fibrosis in 10-15% of the liver.

Response: Porcine METAVIR scores have now been provided for all sections (Suppl Table 2).

- Lines 205-207. METAVIR scores for the adult wild-type pigs (and inclusion of number of controls used) should be provided for result interpretation.

Response: Porcine METAVIR scores for adult wild-type pigs have been provided for result interpretation (line 228).

- Line 208. Only a single pig is listed. Was this pig untreated or chronically undertreated? Was only 1 un/undertreated pig used as a comparison? How consistent are these results (METAVIR stage 4 with HCC tumor development within 12 months) in this model? How many HCC tumors developed, and what was their size? Are CT or ultrasound images available to confirm?

Response: This pig was severely chronically undertreated as untreated animals do not survive long-term. Only one undertreated pig was used as a comparison for humane reasons per our IACUC, and numerous adenomas and one HCC tumor were found within this fibrotic liver. Gross pictures of this liver have been included in Suppl Fig 5. Severely undertreated animals experience significant morbidity, mimicking chronic HT1 disease in human patients, and maintaining multiple animals off NTBC long-term would be unethical, particularly since we have already extensively characterized their phenotype after milder NTBC undertreatment (see Ref 59) along with the known natural history data of HT1 in humans. When chronically undertreated with NTBC, animals will develop METAVIR stage 4 within months (see Ref 59; this has been included in our discussion, line 364), so this is a very consistent finding in the model. Development of HCC within a year requires more severe undertreatment. Therefore, in the context of this study we see the development of stage 4 fibrosis between 3-6 months and adenoma formation shortly thereafter with progression to HCC. We did not perform CT or ultrasound for this study but have included gross images of the specimen in Suppl Fig 5 and sectioned (0.5 cm) the entire liver extensively. The sectioning and staining process alone took several weeks to complete.

- Lines 211-213. It is difficult to effectively screen a 1 year old pig liver at necropsy via sectioning. Was CT or other imaging performed to confirm the lack of HCC tumor development in any of these pigs?

Response: We agree with the reviewer that screening a 1 year old pig liver for adenomas and HCC can be difficult. This is why we took great care to section the entire liver into 0.5 cm sections and examined them closely (see Suppl Fig 5). This was painstaking but we needed to be sure that our histology matched our AFP data and our RNAseq data. In the final analysis, we did confirm that in the treated (LV-FAH) animals we have normal AFP levels, no increased gene expression related to inflammation, fibrosis, or HCC development and that our gross and histologic examinations were consistent with these findings. Nevertheless, we have included absence of imaging in our discussion of limitations (line 470).

- Lines 320-323. A reference should be provided for the statement that untreated FAH^{-/-} pigs develop fibrosis within weeks and progress to hepatic adenomas within 6 months and HCC within 1 year.

Response: A reference has been included to our chronic phenotype characterization study, in which demonstrated the development of fibrosis with undertreatment (see Ref 59). Development of HCC within a year occurs with an undertreatment regimen designed specifically for this study to model long-term potential complications of HT-1 as closely as possible, and which was only performed in one pig, whose NTBC cycling schedule is included in the manuscript.

- Line 405. The lack of CT/MRI imaging to extensively profile for HCC or adenoma formation in these pigs should be listed as a limitation of the study. It should also be noted that longer term studies may be required to confirm HCC development has been eliminated as opposed to simply delayed.

Response: We appreciate the reviewer's feedback. We did not use CT/MRI imaging and have included this as a limitation in our discussion (line 470). However, we section each liver into approximately 0.5 cm sections for gross evaluation of tumors and adenomas. We have included these gross images in the supplemental materials (see Suppl Fig 5). We understand that we cannot conclusively prove complete avoidance of HCC over the animal's entire lifespan; however, we feel strongly that the absence of fibrosis/cirrhosis, elevated AFP, or concerning hepatic gene expression changes after > 6 months off NTBC are protective against future tumor development. If fibrosis, etc. has not developed at this point, we would not expect it to develop in the future, as from our previous experience complete NTBC withdrawal leads to rapid acute liver failure while chronic NTBC undertreatment leads to fibrosis in 100% of animals within weeks to a few months.

- Given the limited number of animals studied (n=2 for 1 year), more details regarding the % of untreated/undertreated FAH^{-/-} pigs that develop cirrhosis, adenomas, and HCC within 1 year should be provided. If 100% of untreated/undertreated FAH^{-/-} pigs develop cirrhosis or HCC within 1 year, this is clearly a significant result. But if only 20% do, then the lack of cirrhosis or HCC development in the 2 animals studied could be due to chance.

Response: When chronically undertreated with NTBC, 100% of animals develop cirrhosis within 3 months to 1 year (see Ref 59; this has been included in our discussion, line 364). Although not all historically undertreated animals developed HCC within this timeframe, our study animal did: pig 266 was undertreated and only one animal underwent this treatment regimen for humane reasons per our IACUC. We have shown in previous studies that all untreated and undertreated animals develop irreversible fibrosis and nodular cirrhosis. Some of these animals will also develop adenomas and HCC such as the animal in this study. Previous reports in humans show that 50% of untreated HT-1 patients who enter the chronic form of the disease will develop adenomas and HCC by age 3. In a large animal model, such as the pig, we would have to study dozens of animals who would suffer significant morbidity to determine the exact timing of adenoma and HCC development. This is prohibitive by IACUC standards. Mice, as we discuss in the manuscript, unfortunately do not develop adenomas and HCC in the backdrop of fibrosis and cirrhosis like humans and pigs (see Ref 37, 60). Our data suggests that the absence of fibrosis is protective, since pigs develop HCC in a fibrosis-dependent manner as do humans (see Ref 60-64).

- Line 443. How were FAH^{-/-} pigs produced? Is this the same herd used in previous studies where cirrhosis and HCC tumor development has been demonstrated previously? If so a reference to the paper describing their initial production should be provided.

Response: FAH^{+/-} pigs were produced through somatic cell nuclear transfer and were bred to produce FAH^{-/-} pigs. All pigs used in this and previous studies belong to the same original herd after some outbreeding. We have included references to initial production of the FAH^{+/-} pig, subsequent breeding for production of a FAH^{-/-} herd, and chronic phenotype characterization of the FAH^{-/-} pig model.

- Line 468. Data from pig 158 is not described in the results. Did this pig also develop cirrhosis and HCC tumors within 1 year?

Response: Pig 158 was included to portray the usual NTBC undertreatment regimen that FAH^{-/-} pigs are able to tolerate, which results in fibrosis and cirrhosis. The development of cirrhosis and HCC within a year occurs with a more aggressive undertreatment regimen that only one pig (266) was subjected to for humane reasons.

- Line 479-480. How many untreated FAH^{-/-} and wild type controls were used for this comparison? Were they age matched?

Response: Two wild-type and six historical undertreated controls were used for comparison. They were age-matched. This information has been included in the text (line 555).

- Line 571. It's unclear how many/which samples from which animals were sequenced, and how these samples were collected and stored. How RNA was isolated, library preparation, sequencing information (platform and single vs paired end) is also not provided. Comparisons performed and number of differentially expressed genes for each comparison should also be provided.

Response: RNA sequencing samples were obtained from 12 different areas in the left and right lobes in all animals with the exception of the animal which developed HCC (266), for which samples were taken from left, middle, and right liver for broader representation given the fact that only one animal was sequenced. Samples were taken from two wild-type animals and three FAH^{-/-} animals adequately treated with NTBC, as well as our two long-term experimental animals and 266 as previously discussed. These samples were collected at necropsy and flash frozen. This has been clarified in the text, along with methodology for RNA isolation, library preparation, and sequencing information (beginning at line 668).

- Figure 4 and 5 are hard to read due to the very small text.

Response: We appreciate the reviewer's feedback. We have increased the size of some of the text.

- Figure 5a. Unclear why these genes were chosen to be displayed in this heatmap. Some are cancer or fibrosis-related, but not all and it's unclear the significance of the ones chosen.

Response: Genes displayed in Fig 5a were chosen after extracting median, minimum, and maximum values from the expression data across the samples, removing all genes with median expression less than 25 and minimum expression of 0, calculating a fold change comparing minimum and maximum expression and removing all genes with a fold change less than 10, and extracting remaining genes based upon their expression pattern of trending towards cancer. This brought 25,149 genes to 4,288 genes to 30 genes. Those 30 genes were then mapped to functional categories, finding cancer- and fibrosis-associated genes to be enriched. This has been clarified in the text (line 690).

- Figure 5b. Are these heatmaps based on all expressed genes or DEGs? Some sort of cluster analysis (heatmap, PCA) for all expressed genes would be valuable to confirm global expression is more similar between LV-FAH treated and WT compared to NTBC/untreated pigs.

Response: The genes in Fig 5b are not based on a differential expression analysis but rather are all the genes within the transcriptome that fit our expected pattern, and therefore shows that there are many genes complementing our expectations and that those genes are functionally enriched in categories that make biological sense. We have now included in Fig 5 a PCA showing all genes (n=1260) from quartile 4 after removing artifacts. The PCA supports our claim that transcriptome-wide the wild-type pigs are more similar to the LV-FAH treated pigs than the NTBC pigs.

- Why was RNAseq only performed on 1 of the pigs treated with LV-FAH and maintained for 12 months?

Response: We have now performed and included in the manuscript RNA seq data and analysis on both of the treated animals maintained for 12 months (see Fig 5). This result was the same and consistent with the other long term treated pig.

- Accession numbers for the genomic/RNA-seq data produced do not appear to be provided.

Response: We have now provided accession numbers for these data as described in the data availability section.

Reviewers' Comments:

Reviewer #1:

Remarks to the Author:

The specific comments have been properly addressed. The reviewer acknowledges that, within the limitations of a large animal study (time of study, number of animals etc), no obvious risk of tumorigenesis could be observed. The decision to proceed to clinical studies is in the hand of regulatory bodies and cannot be the basis for the decision to publish the present data. Experiments and data are now sound and will be of high interest in the field.

Reviewer #2:

Remarks to the Author:

The authors tried to convince me to believe their data but did not provide experimental evidence. For the safety concern, PCR in tissue level is not accurate; they could design a FISH experiment to detect the existence of LV in the liver, which does not need to isolate cell types. I do not believe these results support additional clinical programs to treat HT1 patients with safety concerns.

Reviewer #3:

Remarks to the Author:

The authors have thoroughly addressed the majority of my comments and the revised manuscript is significantly improved. However, I have several follow up questions related to the additional RNA-seq methods/analysis that was provided.

- "We also performed a principal component analysis including all genes (n=1260) from quartile 4". It is unclear how genes were ranked to determine what quartile they were in, and what quartile 4 represents. This should be clarified. If the authors are interested in showing that the expression levels from LV-FAH treated pigs are more similar to WT pigs at the transcriptome-wide level, then the provided PCA plot should be based on all genes that passed filtering (4,288 in this case, although filters seem to be unnecessarily strict as discussed below).

- The RNA-seq analysis is a little confusing. The authors state MAP-Rseq was used for RNA-seq analysis, but also state STAR, featureCounts, and edgeR were used. The pipeline should be clarified.

- Related to filtering, it is unclear what "expression less than 25" refers to (i.e. count, TPM, FPKM). Why was such a high cutoff (resulting in elimination of ~80% of the transcriptome) used? Filtering based on expression is typically used to eliminate lowly expressed genes or gene that are not expressed in all samples, which should not represent the majority of genes profiled unless the starting material used for sequencing was degraded/poor quality.

- The statement "extracting remaining genes (n=30) based upon their expression pattern of trending towards cancer" needs clarification. What does this mean exactly? It's important to understand how these genes were selected because the subsequent pathway analyses (which indicates these genes were involved in cancer/fibrosis pathways) is biased if DEGs were selected based on their involvement in cancer. Typically DEGs are selected solely on expression using edgeR or DeSeq2 to identify statistically significant differences in expression (i.e. q-values < 0.05). Why was this approach not used?

- What groups were used to determine fold changes? Authors state genes with a fold change less than 10 (also a very high cutoff, why is such a high cutoff being used?) were removed. Authors also state comparisons were performed between FAH-/- sick pig No, 266 and wild-type (n=2), FAH-/- pigs adequately maintained on NTBC (n=3), and experimentally treated pigs Nos. 166 and 167. With so many different groups, how were fold changes calculated?

- In the Figure 5 legend the statement "Genes displayed are the result of overlapping differential expression analyses across the three comparisons with absolute log₂ fold change > 2 and FDR < 0.05." Indicates the RNA-seq analysis was performed using a different approach than what is described in the methods. These discrepancies need to be corrected.

- Figure 5b, what genes are represented in these heatmaps? They appear to be DEGs when comparing WT vs Sick and WT vs NTBC based on the heatmap titles, but the legend says it is LV-FAH vs NTBC-undertreated and LV-FAH vs NTBC-treated. Both appear to be in contrast with what is written in the methods.

REVIEWER COMMENTS

Reviewer #1 (Remarks to the Author):

The specific comments have been properly addressed. The reviewer acknowledges that, within the limitations of a large animal study (time of study, number of animals etc), no obvious risk of tumorigenesis could be observed. The decision to proceed to clinical studies is in the hand of regulatory bodies and cannot be the basis for the decision to publish the present data. Experiments and data are now sound and will be of high interest in the field.

Response: We appreciate the reviewer's comments and decision.

Reviewer #2 (Remarks to the Author):

The authors tried to convince me to believe their data but did not provide experimental evidence. For the safety concern, PCR in tissue level is not accurate; they could design a FISH experiment to detect the existence of LV in the liver, which does not need to isolate cell types. I do not believe these results support additional clinical programs to treat HT1 patients with safety concerns.

Response:

We have appreciated the excellent feedback from Reviewer 2 and believe this feedback has strengthened our submission. However, regarding this last topic we have used the exact FDA standards to address the issues of biodistribution and integration. I refer the editor and reviewer to FDA guidance document docket number FDA-2018-D-2173 issued by the Center for Biologics Evaluation and Research (CEBR) Section IV Part B, Part 2 page 10- Considerations for Preclinical Study Design to Assess Biodistribution and Persistence of Gene Therapy Product: Tissue Collection and Analysis: "Use a quantitative, sensitive assay such as quantitative PCR, to analyze the samples for vector sequences."

We have been working with the FDA to design our IND-enabling studies. The data presented here is consistent with their written and verbal guidance and the FDA supports this approach. The data analysis in this study far exceeds the standard guidance already as we analyze all tissues with PCR for biodistribution and VCN, as well as NGS with subsequent bioinformatics to examine all integration events. We acknowledge that other cell types in the liver beyond hepatocytes may be lentiviral integration targets as well. Knowing the exact cell types targeted within in a particular organ is not a part of specific CEBR guidance, which this project and its data has been fully evaluated by. In addition, we do not believe that knowledge that other hepatic cells sustain lentiviral integration would alter the perceived safety profile of this profile. Targeting of a Kupffer cell, for example, would be unlikely to lead to integration-related adverse events if integration within hepatocytes with the subsequent stress of clonal expansion did not lead to an adverse event. While FISH or other studies might further corroborate our data and provide some extra granularity with regard to non-hepatocyte integration, we feel this data would be redundant and beyond the needs of this submission. A more in-depth discussion to this effect has been included in the limitations section of the manuscript. It is our sincerest hope knowing that we have followed strictly the guidance set forth by the FDA that this will satisfy the concerns of this reviewer and the editors.

Reviewer #3 (Remarks to the Author):

The authors have thoroughly addressed the majority of my comments and the revised manuscript is significantly improved. However, I have several follow up questions related to the additional RNA-seq methods/analysis that was provided.

Response: We appreciate the reviewer's comments and remarks and their attention to this review. We have included point-by-point responses to the reviewer's questions below and have updated our manuscript to include the necessary clarifications related to these comments.

- "We also performed a principal component analysis including all genes (n=1260) from quartile 4". It is unclear how genes were ranked to determine what quartile they were in, and what quartile 4 represents. This should be clarified. If the authors are interested in showing that the expression levels from LV-FAH treated pigs are more similar to WT pigs at the transcriptome-wide level, then the provided PCA plot should be based on all genes that passed filtering (4,288 in this case, although filters seem to be unnecessarily strict as discussed below).

Response: The ranking was based on normalized expression; specifically, quartile 1 includes the genes with the lowest expression that met our filtering criteria, while the genes in quartile 4 were those meeting our filtering criteria with the highest overall expression. Our definition of quartile 4 genes has now been included in the text. We only included genes from quartile 4 within the PCA to avoid any biases from lowly expressed genes. Our filtering is strict to ensure our findings are legitimate even after taking conservative measures.

- The RNA-seq analysis is a little confusing. The authors state MAP-Rseq was used for RNA-seq analysis, but also state STAR, featureCounts, and edgeR were used. The pipeline should be clarified.

Response: MAP-Rseq wraps around the RNASeq software mentioned in the methods section, so samples can be processed in parallel all at once. Our methods have been updated to reflect this.

- Related to filtering, it is unclear what "expression less than 25" refers to (i.e. count, TPM, FPKM). Why was such a high cutoff (resulting in elimination of ~80% of the transcriptome) used? Filtering based on expression is typically used to eliminate lowly expressed genes or gene that are not expressed in all samples, which should not represent the majority of genes profiled unless the starting material used for sequencing was degraded/poor quality.

Response: Genes were filtered and removed based on three criteria. Criteria #1: a gene was removed if any sample had a raw read count of 0 for it; criteria #2: a gene was removed if the median level of expression for it was less than 25 across all samples; and criteria #3: a gene was removed if the sample with the highest raw read count for it was less than 10 times the raw read count from the sample with the lowest read count. This strategy was not used because of issues with the starting material. Rather, it was used to ensure the genes evaluated for biological significance were reliably expressed, and the levels of expression were indeed variable across our samples. We have clarified this in the text.

- The statement "extracting remaining genes (n=30) based upon their expression pattern of trending towards cancer" needs clarification. What does this mean exactly? It's important to understand how these genes were selected because the subsequent pathway analyses (which indicates these genes were involved in cancer/fibrosis pathways) is biased if DEGs were selected based on their involvement in cancer. Typically DEGs are selected solely on expression using edgeR or DeSeq2 to identify statistically significant differences in expression (i.e. q-values < 0.05). Why was this approach not used?

Response: Due to the small sample size, we did not use hard cutoffs like a q-value of 0.05. Instead, we looked at the pattern of normalized expression values, i.e., which genes did we see continually become more abundant and which genes continually became less abundant when comparing across sample conditions. The 30 genes mentioned were the genes that held a consistent pattern when the conditions trended towards Cancer. Later in our study, when we performed the subsequent pathway analyses, finding that those genes were known to be involved in Cancer confirms that our findings make biological sense.

- What groups were used to determine fold changes? Authors state genes with a fold change less than 10 (also a very high cutoff, why is such a high cutoff being used?) were removed. Authors also state comparisons were performed between FAH-/- sick pig No. 266 and wild-type (n=2), FAH-/- pigs adequately maintained on NTBC (n=3), and experimentally treated pigs Nos. 166 and 167. With so many different groups, how were fold changes calculated?

Response: As described above, the fold-change general filter was not based on sample type groupings. Instead, it was used to remove genes with little variation across our samples by comparing the sample with the highest raw read count to the sample with the lowest raw read count for that gene. For the downstream differential expression analyses that compared sample types, edgeR was used to calculate log₂FC and FDR values. Many of our comparisons are from three-ways differential expression analyses, meaning we calculate log₂FC and FDR values for A vs B, A vs C, and B vs C.

- In the Figure 5 legend the statement "Genes displayed are the result of overlapping differential expression analyses across the three comparisons with absolute log₂ fold change > 2 and FDR < 0.05." Indicates the RNA-seq analysis was performed using a different approach than what is described in the methods. These discrepancies need to be corrected.

Response: This is correct. For the sample type comparisons, we performed differential expression analyses with edgeR and used log₂FC and FDR values to determine which genes to include in the heatmaps and plots from the manuscript. We have clarified this in the methods.

- Figure 5b, what genes are represented in these heatmaps? They appear to be DEGs when comparing WT vs Sick and WT vs NTBC based on the heatmap titles, but the legend says it is LV-FAH vs NTBC-undertreated and LV-FAH vs NTBC-treated. Both appear to be in contrast with what is written in the methods.

Response: This is correct. The DEGs visualized in those heatmaps are from the comparison described on each plot's title. We included the LV-FAH samples in those heatmaps so the reader can see how those samples correlate with WT and the specified condition (Sick or NTBC). For the WT vs Sick heatmap, it shows the LV-FAH 6 month sample is similar to the Sick samples, and the LV-FAH 12 month sample is similar to the WT samples. Also, for the WT vs NTBC heatmap, it shows the LV-FAH 6 month sample is similar to the NTBC samples, and the LV-FAH 12 month sample is similar to the WT samples.

Reviewers' Comments:

Reviewer #3:

Remarks to the Author:

The authors have addressed all of my questions regarding the RNA-seq analysis. I have no additional questions at this time, and recommend the manuscript be accepted for publication.

REVIEWERS' COMMENTS

Reviewer #3 (Remarks to the Author):

The authors have addressed all of my questions regarding the RNA-seq analysis. I have no additional questions at this time, and recommend the manuscript be accepted for publication.

Response: We thank the reviewer for their comments.